# CD8[+] T cell self-tolerance permits responsiveness but limits tissue damage

Emily N Truckenbrod[1], Kristina S Burrack[1†], Todd P Knutson[2], Henrique Borges da Silva[1‡], Katharine E Block[1], Stephen D O'Flanagan[1], Katie R Stagliano[1§], Arthur A Hurwitz[1#], Ross B Fulton[1¶], Kristin R Renkema[1**]*, Stephen C Jameson[1]*

[1]Center for Immunology, University of Minnesota, Saint Paul, United States; [2]Minnesota Supercomputing Institute, University of Minnesota, Saint Paul, United States

*For correspondence:
renkemak@gvsu.edu (KRR);
james024@umn.edu (SCJ)

Present address: †Hennepin Healthcare Research Institute, Saint Paul, United States; ‡Mayo Clinic Arizona, Phoenix, United States; §NIAID, NIH, Rockville, United States; #AgenTus Therapeutics, Inc, Boston, United States; ¶iFiBiO Therapeutics, Rockville, United States; **Grand Valley State University, Allendale, United States

**Abstract** Self-specific CD8[+]T cells can escape clonal deletion, but the properties and capabilities of such cells in a physiological setting are unclear. We characterized polyclonal CD8[+] T cells specific for the melanocyte antigen tyrosinase-related protein 2 (Trp2) in mice expressing or lacking this enzyme (due to deficiency in *Dct*, which encodes Trp2). Phenotypic and gene expression profiles of pre-immune Trp2/K[b]-specific cells were similar; the size of this population was only slightly reduced in wild-type (WT) compared to *Dct*-deficient (*Dct*[-/-]) mice. Despite comparable initial responses to Trp2 immunization, WT Trp2/K[b]-specific cells showed blunted expansion and less readily differentiated into a CD25[+]proliferative population. Functional self-tolerance clearly emerged when assessing immunopathology: adoptively transferred WT Trp2/K[b]-specific cells mediated vitiligo much less efficiently. Hence, CD8[+] T cell self-specificity is poorly predicted by precursor frequency, phenotype, or even initial responsiveness, while deficient activation-induced CD25 expression and other gene expression characteristics may help to identify functionally tolerant cells.

## Introduction

Accurate discrimination between harmful (pathogens, toxins, cancerous cells) and non-harmful entities (self, innocuous environmental components, non-pathogenic microbes) underlies effective functioning of the immune system. Understanding the mechanisms that normally enforce immunological tolerance to self is a prerequisite for safely and effectively manipulating the immune system to therapeutically induce or break self-tolerance.

Tolerance can be mediated by the clonal deletion of developing self-reactive T cells (*Hogquist et al., 2005*; *Kappler et al., 1987*). Largely based on studies in transgenic mouse models, this process has long been regarded as common and highly efficient (*Palmer, 2003*). However, recent studies have revealed that thymic clonal deletion is less effective than previously thought (*Richards et al., 2016*). Self-reactive CD8[+] T cells have been shown to escape negative selection in mice (*Bouneaud et al., 2000*; *Zehn and Bevan, 2006*), with one group proposing that up to 4% of peripheral CD8[+] T cells are self-specific (*Richards et al., 2015*). Furthermore, studies in humans indicated that precursor frequencies of blood CD8[+] T cells specific for certain self-peptides were comparable to those demonstrated for foreign peptides (*Yu et al., 2015*) and suggested that such cells might be capable of overt autoreactivity if suitably stimulated (*Maeda et al., 2014*).

Aside from clonal deletion, tolerance mechanisms include ignorance of antigen, suppression by regulatory T cells (Tregs), and induction of a functionally unresponsive or hyporesponsive anergic state (*Mescher et al., 2007*; *Mueller, 2010*; *Redmond and Sherman, 2005*; *Schietinger and Greenberg, 2014*). However, different models have produced conflicting evidence regarding the contribution of each of these mechanisms and whether non-deletional CD8[+] T cell tolerance is an intrinsic

property of tolerant cells (*Yu et al., 2015*) or dependent on restraint by Tregs (*Maeda et al., 2014*; *Richards et al., 2015*). It is also unclear how the presence and reactivity of self-specific CD8$^+$ T cells relates to their ability to drive immunopathology. The majority of commonly used mouse models of tolerance have the drawbacks of relying on T cell receptor (TCR) transgenic animals that may not recapitulate normal physiology or utilizing in vitro analyses for characterization of functionality.

Our studies are intended to provide a better understanding of non-deletional CD8$^+$ T cell tolerance by utilizing a more physiological and translationally relevant mouse model in which an epitope from the melanocyte differentiation enzyme tyrosinase-related protein 2 (Trp2) is recognized by CD8$^+$ T cells as either self or foreign. Trp2, an enzyme involved in melanin biosynthesis encoded by the dopachrome tautomerase (*Dct*) gene, is normally expressed by melanocytes in the skin in both humans and C57BL/6 mice and is overexpressed by many melanomas (*Avogadri et al., 2016*; *Wang et al., 1996*). Using wild-type (WT) mice and a novel *Dct*-deficient (*Dct$^{-/-}$*) strain, we compared responses to Trp2$_{180-188}$/K$^b$ (Trp2/K$^b$) as a self- versus foreign antigen. This model is relevant to human health, as Trp2 is a common target in cancer immunotherapy directed against melanoma (*Cho and Celis, 2009*; *Liu et al., 2014*; *Parkhurst et al., 1998*), and Trp2/K$^b$-specific responses can be induced in WT mice with vigorous priming approaches (*Bowne et al., 1999*; *Byrne et al., 2011*; *Cho and Celis, 2009*). Instead of utilizing TCR transgenic models, we focus on the polyclonal Trp2/K$^b$-specific CD8$^+$ T cell repertoire to maximize applicability to normal physiology.

Here, we show that tolerance among Trp2/K$^b$-specific CD8$^+$ T cells is manifest primarily at the level of minimizing overt autoimmunity; differences in the size and initial Trp2 responsiveness of the precursor pool in WT and *Dct$^{-/-}$* mice were relatively modest, although there was evidence for 'pruning' of cells with the highest avidity Trp2/K$^b$-specific TCRs from the WT precursor population. The underlying tolerance mechanism does not depend on cell-extrinsic regulation but rather correlates with a cell-intrinsic failure of WT Trp2/K$^b$-specific CD8$^+$ T cells to sustain optimal proliferation. However, while differences in the responsiveness of WT and *Dct$^{-/-}$* cells to Trp2 immunization were mostly subtle, a notable difference emerged when the cells were assessed for their ability to provoke autoimmune vitiligo: cells primed in *Dct$^{-/-}$* mice were much more effective than those primed in WT animals. Accordingly, we conclude that tolerance in this setting does not lead to marked changes in the presence, phenotype, or initial reactivity of Trp2/K$^b$-specific CD8$^+$ T cells but limits these cells' capacity for overt autoreactivity. Moreover, our polyclonal model reveals that certain characteristics of Trp2/K$^b$-responsive effector cells—reduced CD25 expression and impaired differentiation into a highly proliferative subpopulation—correlate with functional tolerance of a T cell population, providing a framework for future characterization of self-specific CD8$^+$ T cells.

## Results

### The pre-immune population of Trp2/K$^b$-specific cells is similarly sized and appears naïve in WT and *Dct$^{-/-}$* mice

Clonal deletion is a well-studied tolerance mechanism that may result in the near-total culling of self-specific cells or a reduction in the number and apparent TCR affinity of surviving self-specific cells (*Bouneaud et al., 2000*; *Cheng and Anderson, 2018*; *Enouz et al., 2012*; *Hogquist et al., 2005*; *Zehn and Bevan, 2006*). We assessed the number and phenotype of Trp2/K$^b$-specific cells in pre-immune (naïve) mice to examine deletional central tolerance in our model. For mice in which Trp2 would not be a self-antigen, we used a novel *Dct$^{-/-}$* strain that carries a large deletion encompassing the exon (exon 2) encoding Trp2$_{180-188}$ (*Figure 1—figure supplement 1*, *Figure 1—source data 2* and *3*), unlike a previously described *Dct*-targeted strain that retains the coding sequence for that epitope (*Guyonneau et al., 2004*). We performed tetramer enrichment from the spleen and lymph nodes of pre-immune WT and *Dct$^{-/-}$* mice to quantify the number of Trp2/K$^b$-specific CD8$^+$ T cell precursors (*Figure 1A*, *Figure 1—figure supplement 2A*). While on average we identified more antigen-specific cells in *Dct$^{-/-}$* mice (~1.4-fold increased), we nevertheless found relatively large numbers of Trp2/K$^b$-specific CD8$^+$ T cells (>1500) in both strains, evidence that most Trp2/K$^b$-specific cells escape thymic or peripheral clonal deletion in WT mice.

In some systems, T cells bearing TCRs with low-affinity for self-antigens avoid deletion; low-affinity TCRs can often be identified by reduced peptide/MHC tetramer binding to these cells (*Bouneaud et al., 2000*; *Daniels and Jameson, 2000*; *Daniels et al., 2006*; *Enouz et al., 2012*;

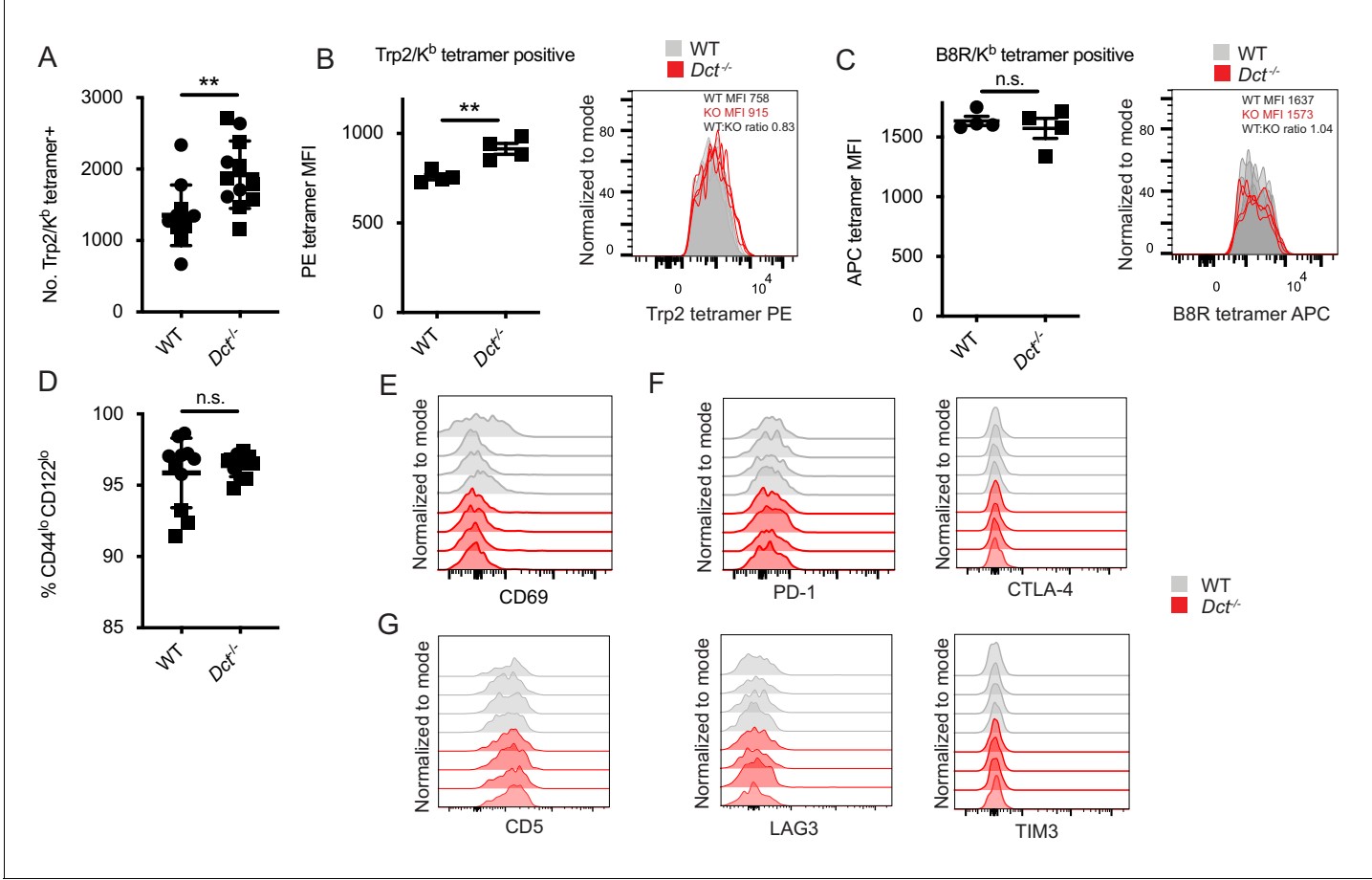

**Figure 1.** Trp2/K$^b$-specific CD8$^+$ T cells in pre-immune wild-type (WT) and *Dct*$^{-/-}$ mice share a naïve phenotype while showing modest differences in frequency and tetramer staining. (**A**) Tetramer enrichment was performed to enumerate Trp2/K$^b$-specific CD8$^+$ T cells per mouse. Median tetramer fluorescence intensity (MFI) was used to estimate the avidity of enriched Trp2/K$^b$-specific (**B**) or B8R/K$^b$-specific cells (**C**). (**D**) CD44/CD122 expression of Trp2/K$^b$-specific cells. (**E**) CD69 expression of Trp2/Kb-specific cells. (**F**) PD-1, CTLA-4, LAG3, and TIM3 expression of Trp2/K$^b$-specific cells. (**G**) CD5 expression of Trp2/K$^b$-specific cells. Data are compiled from three independent experiments in A and D. Individual experiments are shown in B and C; results are representative of other experiments. The graphs in (**E, F, and G**) represent individual experiments with four mice per group. Squares indicate male animals. **p<0.01 by unpaired t test.

The online version of this article includes the following source data and figure supplement(s) for figure 1:

Source data 1. Data file related to *Figure 1*.

Source data 2. Primers used to screen prospective clones for homologous recombination of the *Dct* knockout construct.

Source data 3. Primers used to screen mice to confirm deletion of the *Dct* gene.

Figure supplement 1. Generation of *Dct*$^{-/-}$ mouse model by deleting exons 2 through 6.

Figure supplement 2. Additional analysis of the Trp2/K$^b$-specific population and cells specific for other melanocyte epitopes in pre-immune mice.

*Hogquist et al., 2005*; *Zehn and Bevan, 2006*; *Zehn et al., 2009*). We compared the Trp2/K$^b$ tetramer median fluorescence intensity (MFI) in pre-immune WT and *Dct*$^{-/-}$ mice. The average Trp2/K$^b$ tetramer staining was higher on *Dct*$^{-/-}$ cells, but the MFI largely overlapped between the two populations (*Figure 1B*), suggesting that the range of TCR avidities did not markedly differ between the Trp2/K$^b$-specific pools. Indeed, the tetramer staining difference we observed (a WT:*Dct*$^{-/-}$ tetramer MFI ratio of ~0.8) is more subtle than that noted in a previous study using a transgenic mouse model, which reported a tetramer ratio of ~0.35 between mice with versus without self-antigen expression (*Bouneaud et al., 2000*). As a control to make sure there were no staining differences between the strains, we also assessed the avidity of cells specific for an irrelevant foreign epitope—B8R/K$^b$ from vaccinia virus—in WT and *Dct*$^{-/-}$ mice; the tetramer MFI of B8R/K$^b$-specific cells was comparable between the strains as expected (*Figure 1C*).

We also examined the phenotype of Trp2/$K^b$-specific cells in pre-immune WT and $Dct^{-/-}$ mice. No consistent differences in the expression of activation/memory markers (CD69, CD44, CD122) or anergy/exhaustion markers (PD-1, LAG3, CTLA-4, TIM3) were identified between $Dct^{-/-}$ and WT Trp2/$K^b$-specific cells (*Figure 1D–F*, *Figure 1—figure supplement 2B–D*). The majority of cells exhibited low expression of the memory markers CD44 and CD122, and anergy/exhaustion marker expression was low in both populations. CD5 can indicate self-antigen recognition (*Azzam et al., 1998*; *Fulton et al., 2015*), but we did not detect significant differences in expression between the groups (*Figure 1G*). RNAseq analysis of Trp2/$K^b$ tetramer-binding cells isolated from pre-immune mice by fluorescence-activated cell sorting (FACS) showed no consistent differences in gene expression related to their derivation from WT versus $Dct^{-/-}$ mice (*Figure 1—figure supplement 2E*), although this does not rule out the possibility of epigenetic differences between the populations.

To ensure our findings were not unique to Trp2$_{180}$/$K^b$-specific cells, we used tetramer enrichment to isolate CD8$^+$ T cells specific for other skin antigens—a distinct Trp2 epitope (Trp2$_{363}$/$D^b$) and a tyrosinase-related protein 1 epitope (Trp1$_{455}$/$D^b$)—in mice expressing or lacking these antigens. We were able to identify cells with these specificities present at numbers similar to slightly less in mice expressing antigen relative to mutant mice (*Figure 1—figure supplement 2F*). This suggests that CD8$^+$ T cells specific for other melanocyte self-epitopes also largely escape clonal deletion.

Hence, although we identified some minor differences between Trp2/$K^b$-specific cells from WT versus $Dct^{-/-}$ mice, these pre-immune populations generally resembled each other in number, phenotype, and gene expression, arguing against extensive clonal deletion or overt steady-state anergy induction as the mechanisms dictating tolerance to this antigen. These findings resonate with studies in humans, which have shown that the precursor frequency and average peptide/MHC tetramer staining intensity are only modestly reduced (or 'pruned') among self-antigen-specific CD8$^+$ T cells (*Yu et al., 2015*) and that self-specific cells can be phenotypically naïve (*Maeda et al., 2014*; *Yu et al., 2015*). Accordingly, these data suggested that analysis of Trp2/$K^b$-specific responses in mice could serve as a useful model for investigating the characteristics and responsiveness of self-specific CD8$^+$ T cells that escape deletional tolerance.

## Differences in the magnitude of the response to Trp2 immunization in WT and $Dct^{-/-}$ mice

It was possible that the lack of substantial clonal deletion or signs of prior activation in WT Trp2/$K^b$-specific cells indicated a form of 'ignorance' toward Trp2—that is, T cells capable of strong responses may simply not have encountered or recognized the self-antigen during normal homeostasis, as has been reported in some transgenic models (*Heath et al., 1992*; *Ohashi et al., 1991*). To investigate this, we challenged WT and $Dct^{-/-}$ mice with Trp2 in an immunogenic context using TriVax, a subunit immunization strategy comprising peptide, agonist anti-CD40 antibody, and poly(I:C) (*Cho and Celis, 2009*). It should be noted that the TriVax approach uses the minimal peptide for priming, which likely excludes antigen-specific Treg involvement. We included B8R peptide in addition to Trp2 peptide in these experiments as an internal control. While WT and $Dct^{-/-}$ mice responded similarly to B8R, WT mice showed a more limited response to Trp2 at an effector time point (day 7) relative to $Dct^{-/-}$ mice (*Figure 2A,B*), ruling out this type of ignorance as the dominant tolerance mechanism. We observed a significantly larger number and frequency of Trp2/$K^b$-specific cells in $Dct^{-/-}$ mice, and the $Dct^{-/-}$ cells exhibited higher apparent Trp2/$K^b$ avidity (as measured by tetramer MFI; *Figure 2C–E*); TCRß MFI was similar between WT and $Dct^{-/-}$ Trp2/$K^b$-specific cells, indicating that the difference in tetramer MFI between the strains did not reflect differential modulation of cell-surface TCR (*Figure 2—figure supplement 1A*). Still, WT Trp2/$K^b$-specific cells expanded >1000-fold ($Dct^{-/-}$ cells expanded ~4000-fold). The WT:$Dct^{-/-}$ tetramer ratio was little changed relative to the pre-immune populations, suggesting the difference in avidity between WT and $Dct^{-/-}$ cells had not been amplified by activation.

Aside from the difference in avidity, we were unable to detect meaningful differences in the expression of activation markers, chemokine receptors, or anergy/exhaustion markers at this time point (*Figure 2—figure supplement 1B–D* and data not shown). The frequency of PD-1$^+$ cells was comparable between WT and $Dct^{-/-}$ Trp2/$K^b$-specific populations (*Figure 2F*), suggesting similar exposure to antigen. We also assessed cytokine production and degranulation following ex vivo Trp2 stimulation at day 7 after TriVax and found that a greater frequency of $Dct^{-/-}$ CD8$^+$ T cells produced IFN-γ, TNF-α, and CD107a (an indicator of degranulation), in approximate proportion to the

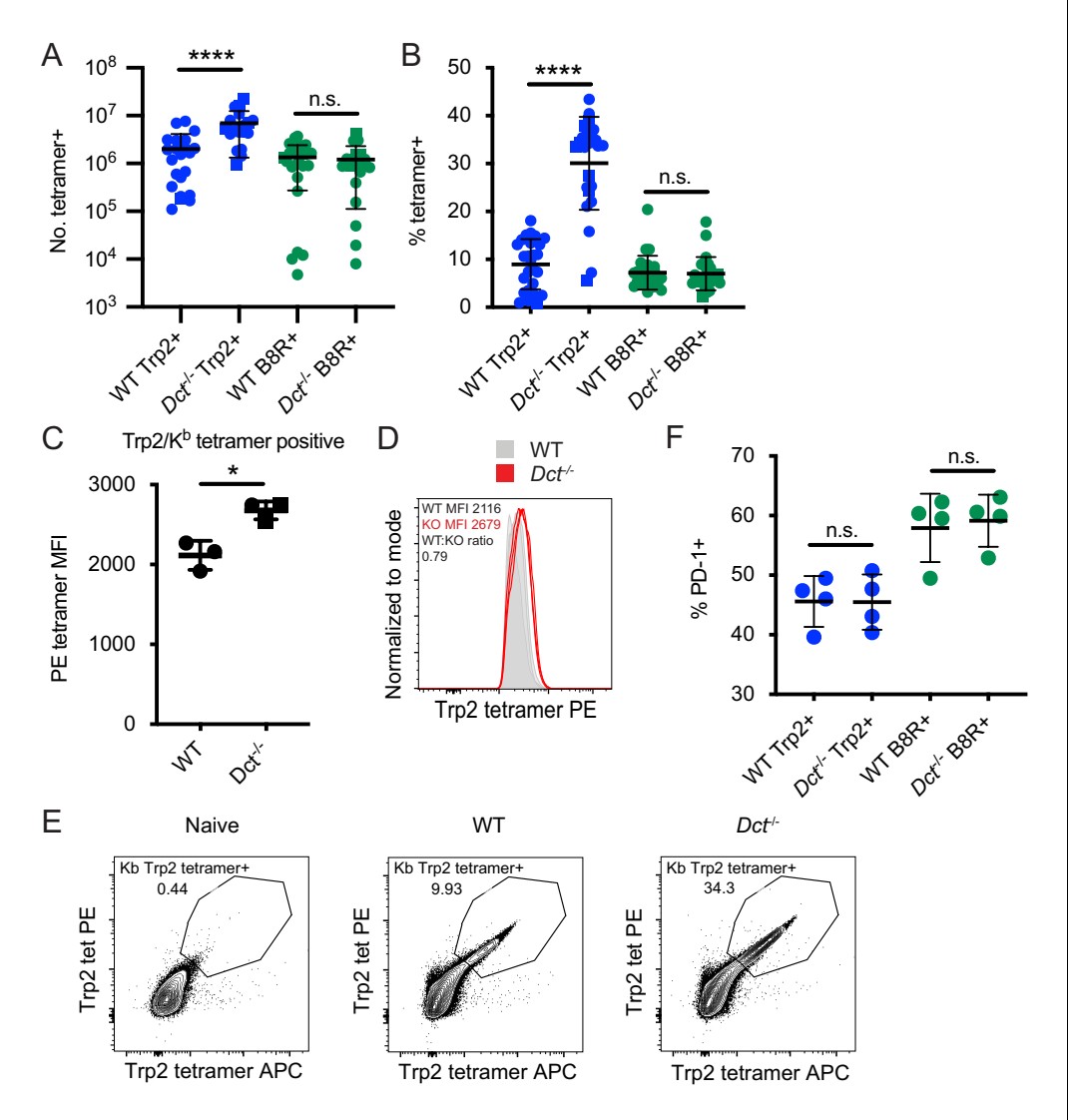

**Figure 2.** Differences in the magnitude of the response to Trp2 immunization in wild-type (WT) and $Dct^{-/-}$ mice. Mice were primed with TriVax (50 µg each of Trp2 and B8R peptides; A–E). The number (A) or percent (B) of splenic Trp2/K$^b$ or B8R/K$^b$-specific cells was assessed at day 7. (C, D) The tetramer fluorescence intensity of splenic Trp2/K$^b$-specific cells was compared. (E) Gating for dual Trp2/K$^b$ tetramer-positive CD8$^+$ (samples were not enriched for Trp2/K$^b$-specific cells). (F) The frequency of the indicated splenic population expressing PD-1 is shown. Data in A and B are compiled from more than three experiments. Data in **C–F** are representative of three or more similar experiments. Squares indicate male animals. *p<0.05, ****p<0.0001 by unpaired t test (**C**) or one-way ANOVA with Sidak's multiple comparisons test (**A, B, F**).

The online version of this article includes the following source data and figure supplement(s) for figure 2:

**Source data 1.** Data file related to *Figure 2*.

**Figure supplement 1.** Additional phenotyping and functional analysis of Trp2/K$^b$-specific cells at day 7 after TriVax.

**Figure supplement 2.** Analysis of Trp2/K$^b$-specific cells in the skin of wild-type (WT) and $Dct^{-/-}$ mice at day 7 after TriVax.

**Figure supplement 3.** Response to infection with *Listeria monocytogenes* strain expressing Trp2 (LmTrp2).

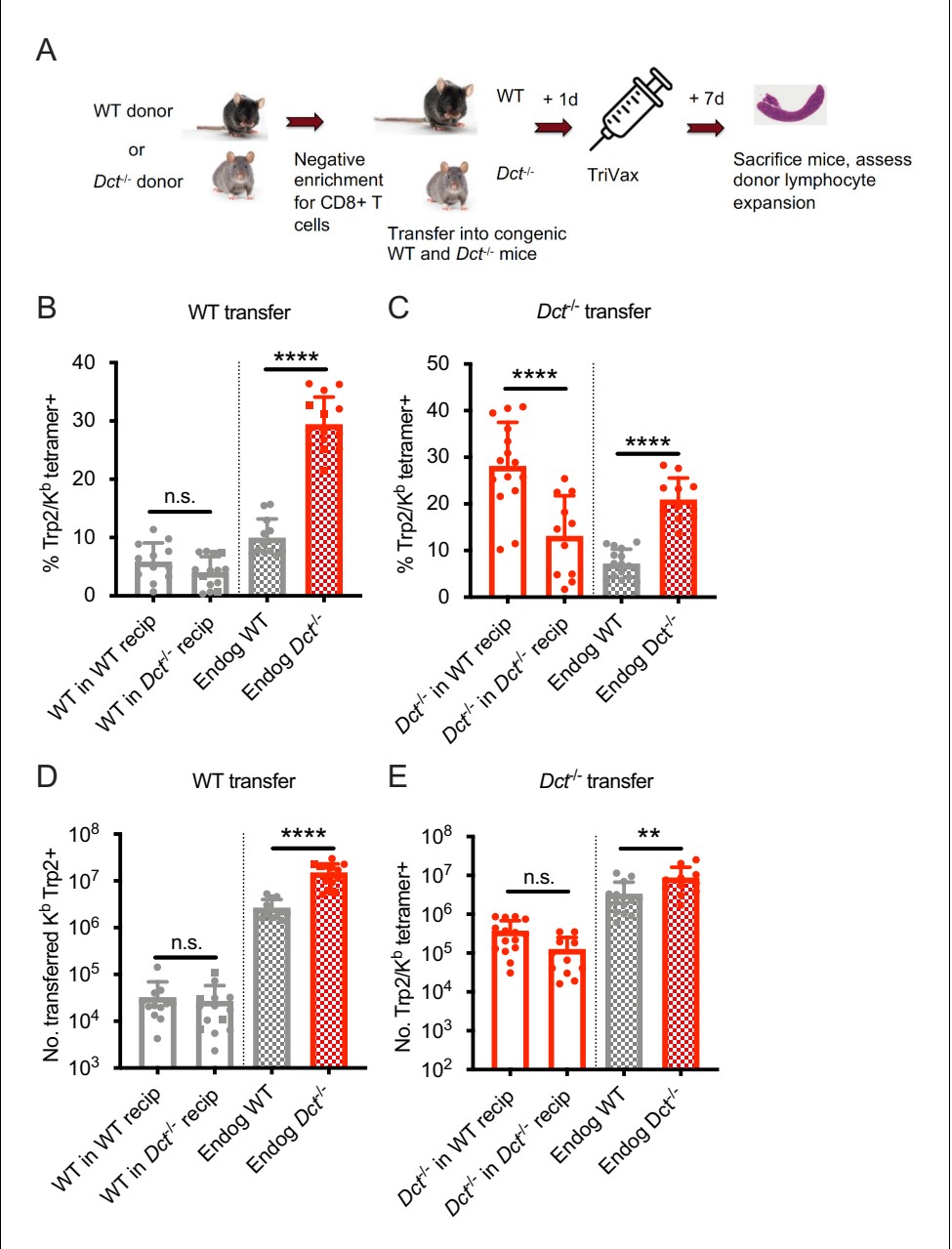

**Figure 3.** Wild-type (WT) Trp2/K$^b$-specific cells exhibit cell-intrinsic tolerance. (**A**) We performed negative enrichment for CD8$^+$ T cells from WT or $Dct^{-/-}$ donors and transferred bulk CD8$^+$ T cells into congenically distinct WT or $Dct^{-/-}$ recipients. One day later, mice were immunized with TriVax (100 μg of Trp2 and B8R peptide). Donor and endogenous cells were collected from the spleens of recipient mice on day 7 following immunization and assessed for the percent (**B, C**) and number (**D, E**) of Trp2/K$^b$-binding cells. Data in B–E were compiled from three or more experiments. Squares indicate male animals. **p<0.05, ***p<0.001, ****p<0.0001 by one-way ANOVA with Sidak's multiple comparisons test. Endog, endogenous; recip, recipients.

The online version of this article includes the following source data and figure supplement(s) for figure 3:

**Source data 1.** Data file related to *Figure 3*.

**Figure supplement 1.** Response to infection with *Listeria monocytogenes* strain expressing Trp2 (LmTrp2) after adoptive transfer.

frequency of Trp2/K$^b$-specific cells (*Figure 2—figure supplement 1E*). Among IFN-γ-producing cells, a slightly larger proportion of *Dct*$^{-/-}$ cells co-produced TNF-α, indicating that Trp2/K$^b$-specific *Dct*$^{-/-}$ cells may have modestly improved polyfunctionality (*Figure 2—figure supplement 1E*).

As the skin is the site of Trp2 expression (in WT mice), we looked at Trp2/K$^b$-specific cells in the skin at day 7 after TriVax to determine whether we would see a more divergent phenotype between WT and *Dct*$^{-/-}$ cells in this location and whether larger numbers of Trp2/K$^b$-specific cells might be attracted to the skin in WT mice, potentially explaining the difference in splenic representation. Interestingly, this was not the case: the number of Trp2/K$^b$-specific cells in the skin was similar between the strains (*Figure 2—figure supplement 2A*). Tetramer MFI was again greater among the antigen-specific *Dct*$^{-/-}$ population; whether the difference in local antigen expression impacted cells' responses was unclear—the proportion expressing CD69 ± CD103 was slightly greater in WT Trp2/K$^b$-specific cells, but we did not detect a clear difference in PD-1 expression (*Figure 2—figure supplement 2B–D*).

Since the Trp2$_{180-188}$ epitope shows suboptimal binding to K$^b$ (*McWilliams et al., 2006*), it was possible that our findings were influenced by the high doses of Trp2 peptide used in the TriVax immunization approach. To explore Trp2 responses in a more physiological context, we infected mice with a recombinant *Listeria monocytogenes* strain expressing Trp2 (LmTrp2) (*Bruhn et al., 2005*) and sacrificed the mice at effector (day 7) and memory (day 45) time points, assessing the percentage and number of Trp2/K$^b$-specific CD8$^+$ T cells and cytokine production in response to ex vivo Trp2 stimulation. Again, the Trp2/K$^b$-specific response was greater in *Dct*$^{-/-}$ mice at both effector and memory time points (*Figure 2—figure supplement 3A–C*). As with TriVax, the frequency of all CD8$^+$ T cells responding to ex vivo Trp2 stimulation with cytokine production (IFN-γ, TNF-α) was larger in *Dct*$^{-/-}$ mice; the percent producing IFN-γ approximated the tetramer-positive population, suggesting that the majority of Trp2/K$^b$-specific cells were able to produce this cytokine in both strains of mice (*Figure 2—figure supplement 3D*). Among IFN-γ-producing cells, those from *Dct*$^{-/-}$ mice tended to produce increased amounts of cytokine on a per-cell basis (as assessed by IFN-γ MFI).

With both the TriVax and LmTrp2 approaches, polyclonal Trp2/K$^b$-specific cells from WT mice showed evidence of tolerance, that is, submaximal responsiveness to Trp2, while those from *Dct*$^{-/-}$ mice mounted a stronger response consistent with a typical response to a foreign antigen. Because cells from *Dct*$^{-/-}$ mice were able to respond robustly in these experiments, the poor responsiveness of WT Trp2/K$^b$-specific cells could not be attributed solely to ignorance or ineffective immunization.

## Tolerance to Trp2/K$^b$ is CD8$^+$ T cell-intrinsic

Both cell-intrinsic and cell-extrinsic mechanisms of CD8$^+$ T cell tolerance have been previously described. Sakaguchi's group (*Maeda et al., 2014*) identified anergic CD8$^+$ T cells specific for melanocyte antigens in healthy human donors and concluded that these cells were restrained by Tregs. In contrast, other groups have shown cell-intrinsic deficits among self-reactive CD8$^+$ T cells. For example, Davis' group *Yu et al., 2015* found human self-antigen-specific T cells to be poorly responsive to antigenic stimulation even in the absence of Tregs, and Greenberg and colleagues (*Schietinger et al., 2012*) showed that tolerant self-reactive murine CD8$^+$ T cells remained tolerant when transferred into new hosts that lacked antigen expression.

Accordingly, we investigated whether cell-intrinsic or cell-extrinsic mechanisms were active in restraining Trp2/K$^b$-specific CD8$^+$ T cells in WT mice. To assess this, we transferred bulk WT polyclonal CD8$^+$ T cells to both WT and *Dct*$^{-/-}$ recipients, then primed the mice with TriVax and examined the effector response at day 7 post-immunization (*Figure 3A, B and D*). Transferred WT Trp2/K$^b$-specific cells did proliferate (~100-fold expansion), albeit to a much lesser degree than endogenous *Dct*$^{-/-}$ Trp2/K$^b$-specific cells. Importantly, their expansion was comparable in both WT and *Dct*$^{-/-}$ recipients (*Figure 3B,D*), suggesting that the WT CD8$^+$ T cells remained hyporesponsive even in an environment where endogenous cells were not tolerant to Trp2, supporting a cell-intrinsic basis for the impaired reactivity of WT Trp2/K$^b$-specific CD8$^+$ T cells. These findings also argue against a model in which the reduced number of splenic Trp2/K$^b$-specific cells arising after priming in WT mice reflects recruitment into sites of self-antigen exposure (e.g., the skin), since similar numbers of transferred WT cells are found in the spleen in WT hosts and *Dct*$^{-/-}$ hosts lacking antigen expression.

We also assessed the performance of *Dct*$^{-/-}$ CD8$^+$ T cells when transferred into *Dct*$^{-/-}$ and WT hosts prior to priming to determine whether they would acquire tolerance in the WT environment

(*Figure 3A*). These Trp2/K$^b$-specific donor cells were able to expand robustly in both *Dct*$^{-/-}$ and WT recipients (*Figure 3C,E*), further demonstrating a lack of extrinsic regulation in the WT environment. *Dct*$^{-/-}$ cells actually performed better in WT recipients than in *Dct*$^{-/-}$ recipients; the basis for this outcome is not clear but could be due to reduced competition by endogenous Trp2/K$^b$-specific cells in WT hosts. Preliminary studies indicated that *Dct*$^{-/-}$ cells still showed strong expansion when the interval between cell transfer and TriVax was extended from 1 day to 1 week, suggesting that these cells did not acquire tolerance characteristics within this timeframe (data not shown).

We conducted similar transfers utilizing LmTrp2 instead of TriVax, again finding evidence of cell-intrinsic tolerance. The transferred cells behaved in accordance with the donors' Trp2 expression rather than that of the recipients: WT cells remained tolerant when primed in *Dct*$^{-/-}$ recipients, while *Dct*$^{-/-}$ cells retained the ability to expand when primed in WT recipients (*Figure 3—figure supplement 1A,B*). Collectively, these data indicate that cell-intrinsic mechanism(s) enforce tolerance among WT Trp2/K$^b$-specific cells.

## WT Trp2/K$^b$-specific cells are capable of an acute response to Trp2

Although the response to Trp2 immunization was weaker in WT versus *Dct*$^{-/-}$ mice, the WT response was still substantial (*Figure 2A*). Studies on T cells with low affinity TCRs have shown a normal initial proliferative response that stalls prematurely relative to the response by high-affinity T cells (*Enouz et al., 2012*; *Ozga et al., 2016*; *Zehn et al., 2009*). Alternatively, it was possible that fewer clones would be recruited into the Trp2 response in WT mice, leading to decreased expansion relative to *Dct*$^{-/-}$ animals from the initiation of an immune response. To distinguish between these possibilities, we studied the expansion kinetics of the Trp2/K$^b$-specific response in WT and *Dct*$^{-/-}$ mice. In order to track early polyclonal responses, TriVax with a higher dose of Trp2 peptide was used in these studies, and tetramer enrichment was used to isolate Trp2/K$^b$-specific cells. Interestingly, WT Trp2/K$^b$-specific cells were capable of an initial response that largely paralleled that shown by their *Dct*$^{-/-}$ counterparts (*Figure 4A*). One day after TriVax immunization, few cells were isolated, likely due to either trapping within the tissues (*Weninger et al., 2001*) or TCR downregulation (*Cai et al., 1997*). Slightly more Trp2/K$^b$-specific cells were identified in WT mice on day 2, while increased numbers of Trp2/K$^b$-specific cells were seen in *Dct*$^{-/-}$ mice on days 3 through 5. By days 6 and 7 after high-dose TriVax immunization, Trp2/K$^b$-specific cells in *Dct*$^{-/-}$ mice outnumbered those in WT mice by an average ratio of 4:1. Although significant, these differences in expansion were modest in comparison with the >1000-fold expansion of Trp2/K$^b$-specific cells in both strains (*Figure 4A*). Preliminary assessment of apoptosis induction (annexin V staining) showed no differences between the strains at 1 or 3 days after TriVax (data not shown).

We also assessed the phenotype of responding Trp2/K$^b$-specific cells acutely after TriVax. With this approach, CD69 did not serve as a reliable indicator of activation due to the type I interferon response induced by poly(I:C) leading to CD69 upregulation (*Shiow et al., 2006*), and widespread CD44 expression was seen in both tetramer-positive and tetramer-negative cells because of the potent inflammatory response unleashed by this method of immunization. Accordingly, we tracked CD25 expression as an indicator of activation. CD25, the high-affinity alpha component of the IL-2 receptor, is upregulated with activation in certain situations (*Valenzuela et al., 2002*) and may enable a stronger effector response by cells expressing it (*Obar et al., 2010*; *Williams et al., 2006*). The proportion of Trp2/K$^b$-specific cells expressing CD25 was significantly greater in *Dct*$^{-/-}$ mice on day 4, and trended higher on days 2 and 3 (*Figure 4B and C*). The CD25 MFI of CD25$^+$ cells was also higher on *Dct*$^{-/-}$ Trp2/K$^b$-specific cells on day 4 (*Figure 4B and C*), suggesting that *Dct*$^{-/-}$ cells expressed more CD25 on a per-cell basis. Once again, *Dct*$^{-/-}$ Trp2/K$^b$-specific cells displayed significantly higher tetramer MFI than WT cells on days 2 through 7, but the avidity differences detected by tetramer staining did not demonstrate a progressive increase with time; the ratio between the WT and *Dct*$^{-/-}$ tetramer MFI transiently dropped at days 4–6, but the ratio at day 7 was similar to that of pre-immune cells (*Figure 4D*, *Figure 4—figure supplement 1A*). As with pre-immune cells, the tetramer MFI of B8R/K$^b$-specific cells was similar between the strains at day 7 (*Figure 4—figure supplement 1B*). Tetramer MFI was highest among the CD25$^+$ subset for both WT and *Dct*$^{-/-}$ cells; the tetramer MFI of CD25$^+$ WT cells was similar to the MFI of the overall tetramer-binding *Dct*$^{-/-}$ population on day 4 (*Figure 4E*).

We also assessed the early response following peptide stimulation alone, since this would be analogous to encountering Trp2 in a non-inflammatory context. We again found the early response

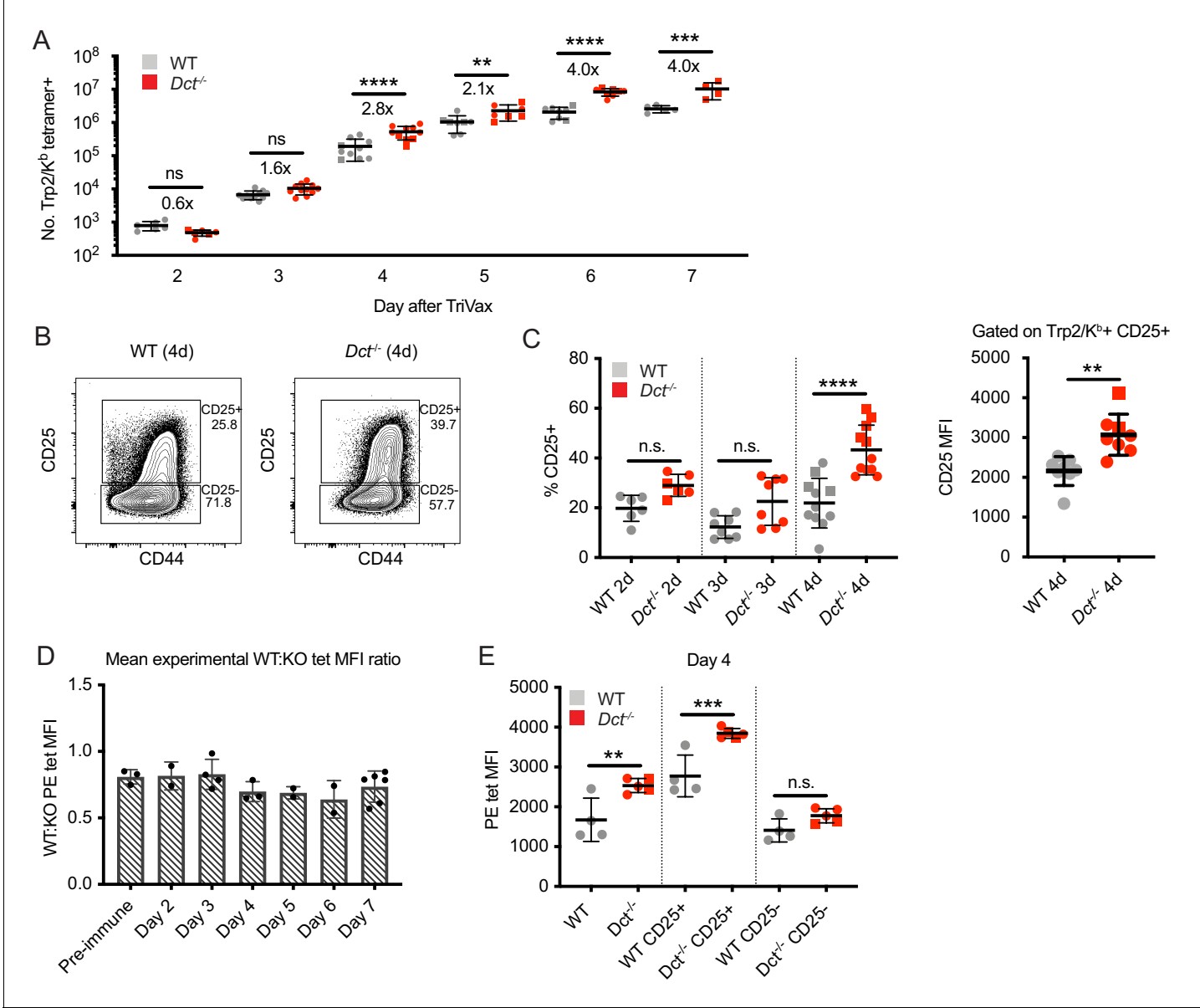

**Figure 4.** Wild-type (WT) Trp2/K$^b$-specific cells are capable of an initial response to Trp2 similar to that of *Dct*$^{-/-}$ cells. WT and *Dct*$^{-/-}$ mice received intravenous injections of TriVax with 200 µg Trp2 peptide. Tetramer enrichment was used to enumerate Trp2/K$^b$-specific cells and assess their phenotype at the indicated time points following immunization (**A–C, E**). The ratio between the mean experimental PE median fluorescence intensity (MFI) of Trp2/K$^b$-specific cells in WT mice relative to *Dct*$^{-/-}$ mice is plotted in D, with each symbol representing one experiment comprising two to five individual mice. Data are compiled from three or more experiments in **A, C,** and **D**. Representative flow plots from 1 day four experiment are shown in **B**, and the same representative day four experiment is shown in **E**. Squares indicate male animals; the dotted line indicates the average naïve precursor frequency from the spleen and lymph nodes. *p<0.05, **p<0.01, ***p<0.001, ****p<0.0001 by one-way ANOVA with Sidak's multiple comparisons test (performed on log-transformed data in [a]).

The online version of this article includes the following source data and figure supplement(s) for figure 4:

**Source data 1.** Data file related to *Figure 4*.

**Figure supplement 1.** Tetramer staining kinetics and response to peptide immunization.

to be similar between WT and *Dct*$^{-/-}$ Trp2/K$^b$-specific cells. The number (***Figure 4—figure supplement 1C***) and phenotype of WT Trp2/K$^b$-specific cells was comparable to that of *Dct*$^{-/-}$ cells on day 1 post-peptide. The activation markers CD44 and CD69 were similarly upregulated in both (***Figure 4—figure supplement 1D,E***), however, responses began to diverge by day 2 after peptide

stimulation, and by day 3 significant differences in number and CD44 expression had emerged, with $Dct^{-/-}$ cells clearly outperforming WT cells (*Figure 4—figure supplement 1C,D*).

These findings indicate that the response to Trp2/K$^b$ in WT and $Dct^{-/-}$ mice follows similar kinetics and magnitude but that expansion in WT animals terminates prematurely. Interestingly, we did not observe a progressive increase in apparent TCR avidity over time among the $Dct^{-/-}$ responder pool relative to WT cells, as might be expected if the subset of $Dct^{-/-}$ T cells with higher avidity TCRs were the only cells capable of responding strongly to Trp2/K$^b$.

## Single-cell sequencing reveals an impaired ability to differentiate into a highly proliferative population early after priming among WT Trp2/K$^b$-specific cells

To better understand the defects in expansion and functionality observed among WT Trp2/K$^b$-specific cells and assess the heterogeneity within this population, we performed single-cell RNA sequencing on Trp2/K$^b$-specific cells from WT and $Dct^{-/-}$ mice at day 7 after TriVax priming. After initial data processing, the WT and $Dct^{-/-}$ datasets were merged (*Stuart et al., 2019*); clusters based on the cells' transcriptomes were generated in an unbiased manner and visualized using uniform manifold approximation and projection (UMAP). Surprisingly, the distribution of WT and $Dct^{-/-}$ cells was quite similar (*Figure 5—figure supplement 1A*); the majority of clusters showed similar frequencies of cells from the two strains, although a slightly larger proportion of $Dct^{-/-}$ cells (14% versus 6% of WT cells) were located in cluster 1 and a slightly smaller proportion in cluster 2 (5% versus 12% of WT cells). We assessed the expression of genes associated with activation, anergy, or exhaustion among these populations and found generally similar patterns of expression between sample groups (*Figure 5—figure supplement 1B*). Aligning with our analysis of anergy/exhaustion marker expression by flow cytometry, we found low expression of these genes in WT (and $Dct^{-/-}$) cells at this effector time point (*Figure 2—figure supplement 1D*; *Figure 5—figure supplement 1B*).

Based on our finding that WT cells began to fall behind $Dct^{-/-}$ cells in number around day 4 after TriVax, it was possible that these populations had diverged earlier after activation. Accordingly, we also performed single-cell sequencing at day 3 after TriVax. At this time point, cells clustered into two major groups separated along the x-axis, each comprised of smaller clusters (*Figure 5A*). Interestingly, over half of the cells (58%) from $Dct^{-/-}$ mice were localized in cluster 0, but this cluster was nearly devoid of WT cells, representing only a small subset (13%) of WT cells (*Figure 5B*). Suspecting that this cluster might contain a more functional subset poorly represented in the WT population, we assessed its characteristics in more detail. Because this cluster made up the majority of Group A (left group), we performed differential gene expression (DE) analysis between the two major groups in the merged dataset: A and B (right group).

Histone genes (e.g., *Hist1h1b*, *Hist1h1e*, *Hist1h1d*, *Hist2h2ac*) were among the most upregulated in Group A compared to Group B; these genes are commonly induced in association with cellular replication (*Mei et al., 2017*). Other genes associated with proliferation, such as *Myc*, *Nolc1*, *Npm1*, and *Ccne2*, were also upregulated in Group A (*Figure 5C*), and cell cycle analysis revealed that the majority of cells in Group A were in stages G2/M or S of the cell cycle (*Figure 5D*). Among cells in cluster 0 (Group A), 76% of $Dct^{-/-}$ cells were in G2/M or S versus 42% of WT cells (*Figure 5E*). Gene set enrichment analysis revealed a strong enrichment of gene sets comprising Myc targets, E2F targets, and genes related to mTORC1 signaling and the G2/M checkpoint (*Figure 5—figure supplement 2A*).

Many of the cells in Group A expressed CD25, with the majority of the remainder located in cluster 2 of Group B (*Figure 5F*); cells in these clusters also showed enrichment for a gene signature associated with IL-2 receptor signaling (*Figure 5—figure supplement 2B*). This aligns with our finding that the frequency of cells expressing CD25 was greater in $Dct^{-/-}$ than WT Trp2/K$^b$-specific effectors (*Figure 4C*). Indeed, Group A cells showed significantly higher expression of certain genes relevant to the IL-2 signaling pathway with known impacts on T cell function, such as *Irf4* and *Myc*; previous work has demonstrated that signaling through the IL-2 receptor is important for sustained *Myc* expression (*Preston et al., 2015*). *Lag3* and *Pdcd1* (PD-1) expression were seen among cells in some clusters, but expression of these markers was higher among $Dct^{-/-}$ cells (*Figure 5—figure supplement 2C*); other anergy and exhaustion markers were not widely expressed in either population, similar to the data from day 7.

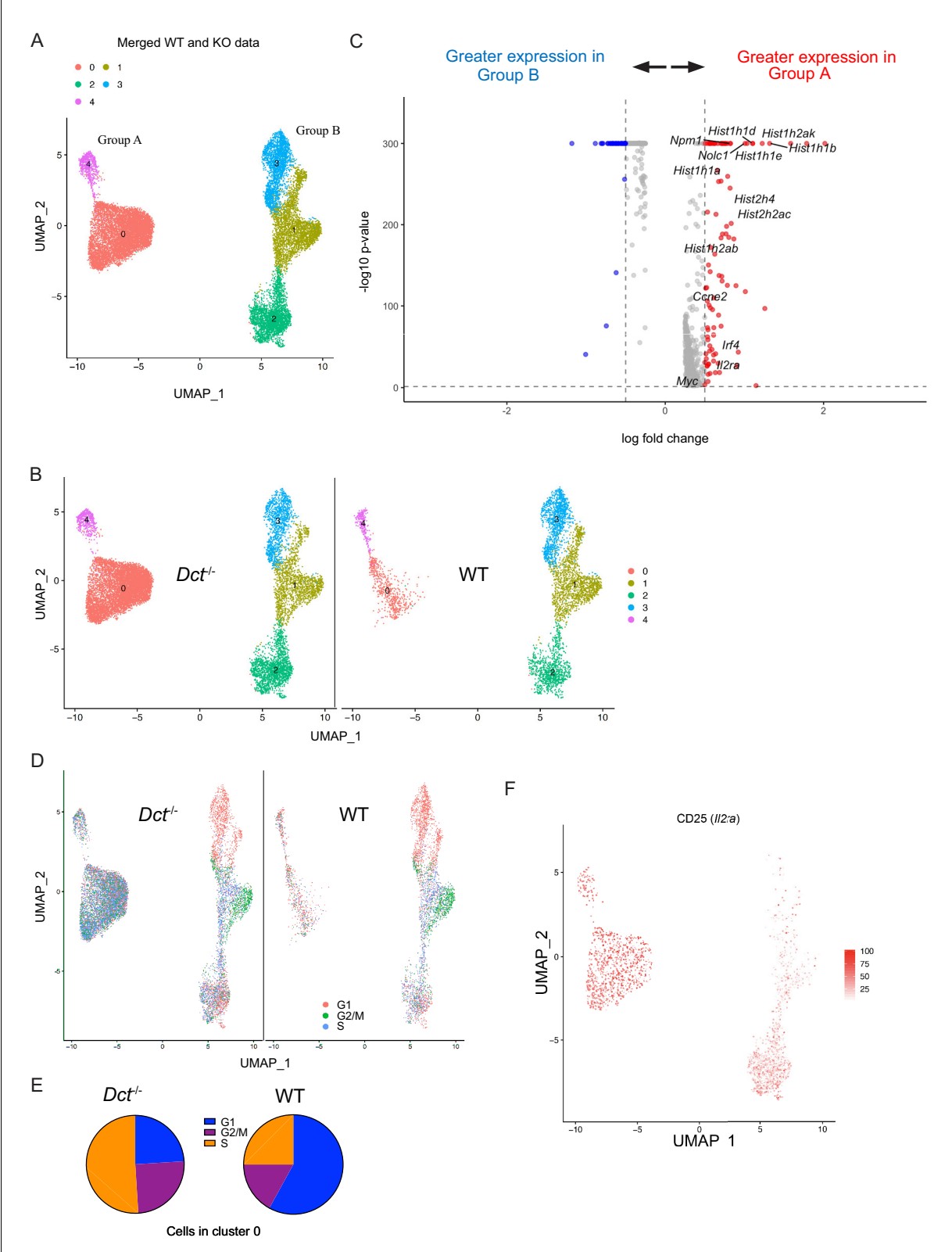

**Figure 5.** Wild-type (WT) Trp2/K$^b$-specific cells show proliferative defects in the early effector phase. Trp2/K$^b$-specific CD8$^+$ T cells were isolated from WT and *Dct*$^{-/-}$ mice on day 3 after TriVax and submitted for scRNA-seq. After initial processing, the WT and *Dct*$^{-/-}$ datasets were merged and further analyzed. (A) Uniform manifold approximation and projection (UMAP) representation of gene expression from merged datasets determined using Seurat; each dot represents one cell. Clusters are indicated by color. (B) Cells from the *Dct*$^{-/-}$ sample are shown on the left and cells from the WT

*Figure 5 continued on next page*

*Figure 5 continued*

sample on the right using the same UMAP (generated from merged data) shown in *Figure 6A*. (C) The most differentially expressed genes between Groups A and B (see *Figure 6A*); histone genes and other genes associated with proliferation are indicated. A positive average log-fold change value indicates higher expression in Group A. (D) Cell cycle analysis indicates the cell cycle phase for each cell on the UMAP plot (merged dataset). (E) Pie charts show the frequencies of cells within cluster 0 in each stage of the cell cycle (left: $Dct^{-/-}$ sample, right: WT sample). (F) Expression of CD25 (*Il2ra*) by cell is indicated on the clusters by color.

The online version of this article includes the following source data and figure supplement(s) for figure 5:

**Source data 1.** Data file related to *Figure 5*.
**Figure supplement 1.** scRNAseq analysis of Trp2/K$^b$-specific cells at day 7 after TriVax.
**Figure supplement 2.** Gene set enrichment analysis (GSEA) of day 3 single-cell data and response of wild-type (WT) and $Dct^{-/-}$ Trp2/K$^b$-specific cells to IL-2C.

Based on the association of IL-2 signaling with $Dct^{-/-}$ Trp2/K$^b$-specific effector cells, we administered IL-2 complex (IL-2 + anti-IL-2 S4B6 antibody) to WT and $Dct^{-/-}$ mice previously primed with Tri-Vax to determine whether this would correct the defective proliferation of the WT Trp2/K$^b$-specific cells. IL-2 complex acts through the β and γ components of the IL-2 receptor, negating the impact of differential CD25 expression. IL-2 complex treatment on day 5 after TriVax or LmTrp2 improved the expansion of both WT and $Dct^{-/-}$ Trp2/K$^b$-specific cells to a similar extent (*Figure 5—figure supplement 2D,E*). Although this treatment did not correct the expansion defect of WT Trp2/K$^b$-specific cells in a selective manner, it did improve their numbers to the level seen among untreated $Dct^{-/-}$ cells, supporting the use of IL-2R-directed therapies in cancer immunotherapy designed to engage tolerant cells (*Moynihan et al., 2016*; *Rosenberg, 2014*; *Waithman et al., 2008*).

Taken as a whole, the RNA-sequencing data suggest that WT Trp2/K$^b$-specific cells are deficient in their ability to form the more proliferative subpopulation that comprises a majority of the $Dct^{-/-}$ Trp2/K$^b$-specific population on day 3 after priming. Nevertheless, proliferation of the WT population was not entirely constrained, since this pool continued to expand over successive days (*Figure 4*).

## WT Trp2/K$^b$-specific cells are inefficient at mediating vitiligo

T cells that escape self-tolerance mechanisms can sometimes elicit autoimmunity. Even CD8$^+$ T cells with very-low-affinity TCRs that avoid deletional tolerance have been found to drive tissue destruction following activation (*Enouz et al., 2012*; *Sabatino et al., 2011*; *Zehn and Bevan, 2006*). Furthermore, vigorous immunization against Trp2 can break tolerance and lead to vitiligo (*Bowne et al., 1999*; *Cho and Celis, 2009*; *Moynihan et al., 2016*). Our data indicated that the proliferative response of Trp2/K$^b$-specific cells was only slightly impaired in WT relative to $Dct^{-/-}$ mice, but the ability of these expanded cells to mediate overt tissue damage, as indicated by autoimmune vitiligo, was unclear.

To investigate this, we primed WT and $Dct^{-/-}$ donors with TriVax, then transferred day 7 effectors to congenically distinct WT recipient mice in parallel and immunized these recipients with TriVax. Equal numbers of $Dct^{-/-}$ and WT cells were transferred to compensate for the reduced response in WT mice. The recipients were treated with dinitrofluorobenzene (DNFB) on the left flank 6 days after transfer as a local inflammatory stimulus (*Haas et al., 1992*; *Mackay et al., 2012*; *Zhang et al., 2009*); vehicle (acetone/olive oil) was applied to the right flank (*Figure 6A*). Analysis of the blood 6 days after transfer and boosting revealed expansion of both types of donor cells; although there was a trend for transferred $Dct^{-/-}$ cells to expand to a greater degree than donor WT cells, the difference was not statistically significant (*Figure 6B*). Recipient mice were subsequently monitored for vitiligo development on a weekly basis and scored using a numeric metric (*Figure 6—figure supplement 1A*).

Recipients of $Dct^{-/-}$ cells developed vitiligo more rapidly and more extensively than mice receiving WT cells, beginning around day 20 after cell transfer (*Figure 6C*). Vitiligo was most frequently initiated at the DNFB-treated site and would often progress over the following weeks to involve the right flank, hair around the eyes, and—in some cases—hair distributed over the body. Vitiligo progressed more rapidly and to a greater extent (higher numeric score) in recipients of $Dct^{-/-}$ cells (*Figure 6D*; *Figure 6—figure supplement 1B,C*), although low-grade vitiligo was observed in some mice receiving WT Trp2/K$^b$-specific effector cells or TriVax and DNFB without cell transfer. It is possible that initial melanocyte destruction mediated by the transferred cells facilitated antigen release

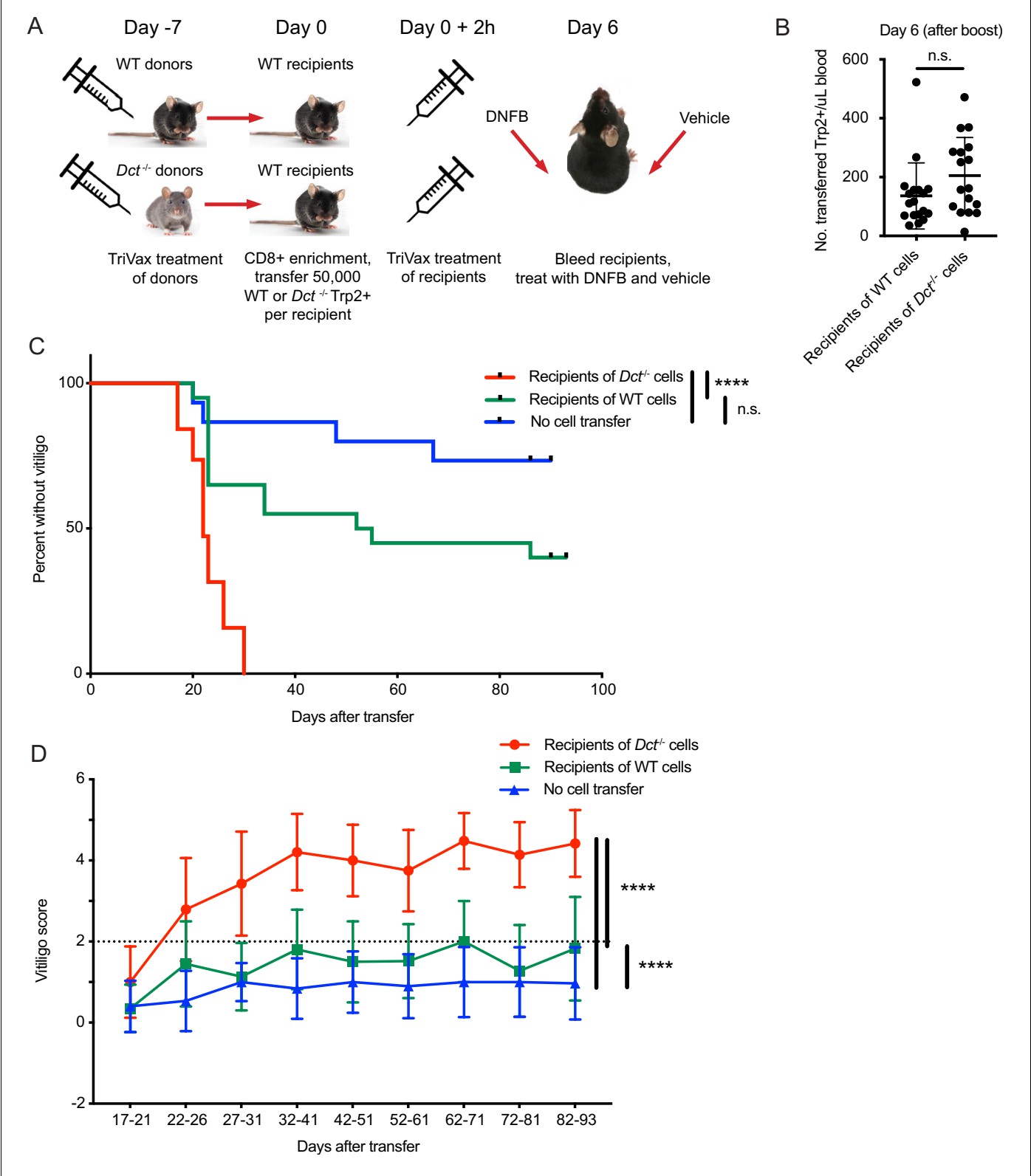

**Figure 6.** Wild-type (WT) Trp2/K$^b$-specific cells are unable to mediate efficient anti-melanocyte activity. (**A**) WT mice were monitored for vitiligo after receiving 50,000 Trp2/K$^b$-specific cells from WT or $Dct^{-/-}$ donors primed with TriVax 7 days prior; recipient mice received TriVax (100 μg Trp2) on the day of transfer and were treated with dinitrofluorobenzene (DNFB) (left flank) 6 days later. No cell transfer controls (not shown in schematic) received TriVax and DNFB but no transferred cells. (**B**) Recipient mice were bled on day 6 after transfer and TriVax; the number of transferred Trp2/K$^b$-positive cells per

*Figure 6 continued on next page*

*Figure 6 continued*

µL blood is shown. (C) Kaplan-Meier curve of vitiligo development; mice were considered to have vitiligo when they first had a vitiligo score of two that was sustained. Mean group vitiligo scores over time are shown in (D), with a dotted line indicating definite vitiligo. Data in C and D are compiled from three experiments with 4–10 mice per group. Data in B are compiled from two experiments with 4–10 mice per group. ****p<0.0001 by unpaired t test (B), log-rank survival analysis (C), or two-way ANOVA followed by Tukey's multiple comparisons test (D).

The online version of this article includes the following source data and figure supplement(s) for figure 6:

**Source data 1.** Data file related to *Figure 6*.
**Figure supplement 1.** Vitiligo scoring metric and correlation between the average vitiligo score and the number of transferred Trp2/K$^b$-specific cells.
**Figure supplement 2.** Vitiligo development and skin infiltration in P14 mice receiving cell transfers from wild-type (WT) or *Dct$^{-/-}$* donors.

and a broadening of the anti-melanocyte response to include endogenous T cells; nevertheless, limited studies using irrelevant TCR transgenic mice (P14) as recipients showed that Trp2/K$^b$-specific donor cells from *Dct$^{-/-}$* mice were still able to induce vitiligo in this setting (***Figure 6—figure supplement 2A,B***).

Vitiligo severity (average vitiligo score) was positively correlated with the number of donor Trp2/K$^b$-specific cells in the blood on day 6 after transfer and TriVax boost (***Figure 6—figure supplement 1D***). This suggests that the enhanced proliferative capacity of *Dct$^{-/-}$* cells was a factor in their superior ability to induce vitiligo, although it does not rule out additional qualitative differences between the WT and *Dct$^{-/-}$* populations.

We also assessed the number and phenotype of transferred Trp2/K$^b$-specific cells in the skin of recipient P14 mice, which lacked a large population of endogenous antigen-specific cells. At day 11 or 12 after cell transfer, we identified similar numbers of cells from both donors in the DNFB-treated (left) flank; interestingly, this number was also similar to the number of cells identified in the vehicle-treated (right) flank (***Figure 6—figure supplement 2C***). The phenotype (PD-1, CD49a expression) of the transferred Trp2/K$^b$-specific cells was also similar across donor group and both flanks (***Figure 6—figure supplement 2D***). As in previous experiments, antigen-specific cells from *Dct$^{-/-}$* donors exhibited higher tetramer MFI; the tetramer MFI on donor cells in the skin was similar to or slightly higher than that observed in the spleen (***Figure 6—figure supplement 2E*** and data not shown). At a memory time point, fewer transferred Trp2/K$^b$-specific cells were identified in the skin, but this reduction was similar in recipients of WT and *Dct$^{-/-}$* cells, as was the number of transferred Trp2/K$^b$-specific cells in the spleen at day 27 after transfer (data not shown). These findings contrasted with the clear difference in vitiligo elicited by *Dct$^{-/-}$* versus WT cells and indicate that differences in skin recruitment and retention are unlikely to account for the distinct disease progression induced by these populations.

In summary, in contrast to the relatively modest differences in the expansion of Trp2/K$^b$ responders in WT and *Dct$^{-/-}$* mice, the ability of these populations to mediate autoimmune damage—melanocyte destruction—was strikingly different.

## Discussion

A number of groups have demonstrated the existence of self-reactive CD8$^+$ and CD4$^+$ T cells in the periphery of mice and healthy human adults (***Anderson et al., 2000***; ***Bloom et al., 1997***; ***Delluc et al., 2010***; ***Maeda et al., 2014***; ***Su et al., 2013***; ***Yu et al., 2015***). In some cases, self-reactive cells display indicators of reduced functionality, revealing them as tolerant and unlikely to cause spontaneous pathology. For example, self-reactive cells are often reported to express inhibitory receptors such as CTLA-4, PD-1, and LAG-3 (***Fife and Bluestone, 2008***; ***Maeda et al., 2014***; ***Nelson et al., 2019***; ***Schietinger et al., 2012***). However, studies in human adults have identified self-reactive cells with a phenotype similar to that of naïve CD8$^+$ T cells specific for foreign antigens (***Yu et al., 2015***); these cells did not display an overtly anergic phenotype but still responded poorly to stimulation. It is important to understand the mechanisms restraining these cells under normal conditions as well as their potential to cause pathology; such knowledge is critical to designing effective therapies to restrain these cells (e.g., to control autoimmune disease) or induce their responses (e.g., for cancer immunotherapy). As described in this report, we developed a polyclonal mouse model for CD8$^+$ T cell self-tolerance, enabling us to define the characteristics of these cells and their reactivity in a physiological setting.

We found that the pre-immune populations of Trp2/K$^b$-specific cells in WT and $Dct^{-/-}$ strains were qualitatively similar, sharing a naïve phenotype and indistinguishable gene expression profile; there were no clear signs of prior antigen exposure among the WT cells. The size of the Trp2/K$^b$-specific precursor pool was only slightly (although significantly) smaller in WT mice, by less than twofold, and the response to Trp2 immunization was substantial in both strains, leading to a >1000-fold expansion of Trp2/K$^b$-specific cells in both WT and $Dct^{-/-}$ mice. Despite these commonalities, the population primed in $Dct^{-/-}$ mice showed greater expansion and elicited more rapid and widespread tissue destruction, read out as vitiligo, after adoptive transfer. Since the enhanced induction of vitiligo by transferred $Dct^{-/-}$ cells did not align with their improved recruitment to the skin, these findings suggest that $Dct^{-/-}$ cells are more efficient at mediating melanocyte destruction within the tissue.

Our adoptive transfer studies showed that the observed restraint in the WT Trp2/K$^b$-specific response did not depend on extrinsic factors but was a cell-intrinsic feature of pre-immune CD8$^+$ T cells. This implies that other cell populations, including CD4$^+$ Tregs or regulatory CD8$^+$ T cells (*Saligrama et al., 2019*) are neither required for nor capable of affecting the responses of tolerant and non-tolerant Trp2/K$^b$-specific cells during priming. These findings also effectively eliminate the possibility that self-antigen presentation during Trp2 priming alters the nature of the immune response. However, while our studies argue that cell-extrinsic regulation is not required for enforcement or maintenance of tolerance by Trp2/K$^b$-specific cells, this does not exclude a potential role for Treg populations in establishing the initial tolerant state in WT mice. It is currently unclear whether tolerance to Trp2 is enforced during thymic development or in the periphery of WT mice: one report suggested that $Dct$ expression is undetectable in thymic mTECs (*Träger et al., 2012*), but Trp2 could be brought into the thymus by dendritic cell populations to induce tolerance in WT animals. The site of tolerance induction was not a focus of the current study, but it will be interesting to determine whether instances of CD8$^+$ T cell self-tolerance correlate with self-antigen expression patterns in the thymus (e.g., AIRE-regulated tissue-specific antigen expression).

Many of the characteristics we report for Trp2/K$^b$-specific CD8$^+$ T cells in WT mice are strongly reminiscent of T cells with low affinity/avidity for antigen (*Bouneaud et al., 2000*; *Enouz et al., 2012*; *Zehn and Bevan, 2006*; *Zehn et al., 2009*), and we did observe modestly higher Trp2/K$^b$ tetramer staining intensity on a subset of $Dct^{-/-}$ cells compared to WT responder cells. The 'pruning' of WT CD8$^+$ T cells with high-avidity Trp2/K$^b$-specific TCRs remains the most likely explanation for this finding. If so, however, the effect of such pruning appears to be remarkably subtle. While analysis of unimmunized mice revealed that the average number and tetramer staining intensity of Trp2/K$^b$-specific CD8$^+$ T cells were significantly lower in WT than $Dct^{-/-}$ animals, the degree of overlap suggests such measures would be unreliable for detecting tolerant cells beyond this optimized experimental model. Furthermore, the relatively consistent WT to $Dct^{-/-}$ tetramer MFI ratio in our time course experiments argues against an imbalanced outgrowth of higher affinity/avidity TCR clones in $Dct^{-/-}$ relative to WT animals (as might have been expected if TCR affinity was the primary factor driving improved proliferation in the $Dct^{-/-}$ population). At a practical level, the relatively modest differences in cell number and tetramer staining between tolerant and non-tolerant cells that we observed in this carefully controlled model system would not be sufficient to accurately predict self-reactivity versus tolerance in a clinical setting. Broadly similar conclusions were drawn from earlier studies of self-tolerance in humans (*Maeda et al., 2014*; *Yu et al., 2015*). This finding highlights the limitations of currently available assays for accurately predicting responsiveness to self-antigens.

Other studies utilizing mouse models have reported that even CD8$^+$ T cells with very-low-affinity/avidity TCRs (including those undetectable by normal peptide/MHC tetramer staining) can provoke overt tissue damage that may reach or exceed the response observed for non-tolerant, high-affinity/avidity cells (*Enouz et al., 2012*; *Sabatino et al., 2011*; *Zehn and Bevan, 2006*). Our data indicate that the opposite can also occur: despite largely overlapping tetramer staining profiles, WT and $Dct^{-/-}$ Trp2/K$^b$-specific cells exhibit markedly different abilities to mediate widespread vitiligo. Hence, the impact of CD8$^+$ T cell tolerance toward some self-antigens only partially limits expansion but can prevent the generation of cells readily capable of potent tissue destruction: tolerance is not a binary state.

Although enumeration, phenotyping, and pre-immune gene expression profiling failed to provide a robust metric for identifying functionally tolerant CD8$^+$ T cells, we were able to delineate an inflection point following priming at which the responses of tolerant and non-tolerant cells diverged. Flow cytometry and single-cell RNA sequencing of Trp2/K$^b$-specific CD8$^+$ T cells soon after priming

demonstrated that WT responders failed to differentiate into a CD25[+], IRF4[+] population (a characteristic of most $Dct^{-/-}$ responder cells) and indicated that WT cells showed poor commitment to sustained proliferation. These combined features may be useful for further defining the responses by self-antigen-specific cells that are or are not capable of overt tissue destruction.

The early effector population of Trp2/K[b]-specific cells demonstrates heterogeneity on a transcriptomic level in both strains. Whereas the majority of $Dct^{-/-}$ cells show a highly proliferative phenotype characterized by active cell cycling and responsiveness to mTOR and Myc, few WT Trp2/K[b]-specific cells fall into this group. The reason(s) underlying the inability of WT cells to optimally engage these important pathways and proliferate efficiently require(s) further investigation but may relate to impaired sensitivity to endogenous IL-2 or other cytokines, the composition of the TCR repertoire, and/or altered TCR signaling. How these or other factors relate to the relative inability of primed WT Trp2/K[b]-specific cells to mediate overt tissue damage is currently unclear, but it will be critical to identify the cellular and molecular mechanisms involved in future studies. Recent studies on dysfunctional tumor-specific or exhausted CD8[+] T cells have shown that epigenetic changes in chromatin accessibility or methylation can maintain such states (*Ghoneim et al., 2017*; *Pauken et al., 2016*; *Philip et al., 2017*), which is one potential explanation for the cell-intrinsic nature of the tolerance seen in our model.

We were able to identify CD8[+] T cells specific for other melanocyte epitopes/antigens in pre-immune mice; these cells had a similar phenotype to WT Trp2/K[b]-specific cells. Accordingly, we predict that our results will apply to other populations of CD8[+] T cells specific for melanocyte and potentially other tissue-restricted antigens. Similar populations of self-specific CD8[+] T cells may exist in humans, and the ability of such cells to respond to self-antigen immunization while not causing autoimmune damage is relevant for understanding the limits of 'breaking' tolerance, for example, for cancer immunotherapy. Indeed, our results align with work examining polyclonal self-antigen-specific cells in human adults (*Yu et al., 2015*) with regard to the phenotype (modestly lower tetramer MFI, lower CD25 expression) and response to cognate peptide (diminished) observed among tolerant cells. Another study examining self-specific CD8[+] T cells (*Maeda et al., 2014*) attributed their restrained responsiveness to Treg-mediated suppression; while we did not detect a cell-extrinsic regulatory mechanism in our studies, it is certainly possible that this mechanism limits the response to some self-antigens.

Our finding that polyclonal melanocyte-specific cells exhibit covert cell-intrinsic tolerance characterized by a partial defect in proliferation and a profound defect in tissue damage has implications for utilizing such cells therapeutically. This model has clear relevance to human physiology and will be useful in exploring methods of correcting the proliferative defects of tolerant cells to more effectively mobilize them in cancer immunotherapy approaches targeting tumor antigens shared with self.

## Materials and methods

### Key resources table

| Reagent type (species) or resource | Designation | Source or reference | Identifiers | Additional information |
|---|---|---|---|---|
| Strain, strain background (*Mus musculus*) | $Dct^{-/-}$ (Exon 2–6$^{-/-}$) | Dr. A. Andy Hurwitz, NCI | | |
| Strain, strain background (*Mus musculus*) | C57BL/6NCrl (C57BL/6) | NCI Charles River | Strain code: 556 | |
| Strain, strain background (*Mus musculus*) | B6.SJL-PtprcaPepcb/BoyCrCrl (CD45.1) | NCI Charles River | Strain code: 564 | |
| Strain, strain background (*Mus musculus*) | P14 | Dr. R. Ahmed, Emory University | | |

*Continued on next page*

*Continued*

| Reagent type (species) or resource | Designation | Source or reference | Identifiers | Additional information |
|---|---|---|---|---|
| Chemical compound, drug | Poly(I:C) | Invivogen | Cat. #: vac-pic | |
| Antibody | InVivoMAb anti-mouse CD40 | BioXCell | Cat. #: BE0016-2; RRID:AB_1107647 | Clone FGK4.5 |
| Peptide, recombinant protein | Trp2 peptide | New England Peptide | | $Trp2_{180-188}$ H2N-SVYDFFVWL-OH |
| Chemical compound, drug | 1-Fluoro-2,4-dinitrobenzene (DNFB) | Sigma Aldrich | Cat. #: D-1529 | |
| Antibody | Anti-mouse CD8a FITC | Tonbo Biosciences | Cat. #: 35-0081; RRID:AB_2621671 | Clone 53-6.7 |
| Antibody | Anti-mouse CD8a vf450 | Tonbo Biosciences | Cat. #: 75-0081; RRID:AB_2621931 | Clone 53-6.7 |
| Antibody | Anti-mouse CD8a PerCP-Cy5.5 | Tonbo Biosciences | Cat. #: 65-0081; RRID:AB_2621882 | Clone 53-6.7 |
| Antibody | Anti-mouse CD8a APCef780 | eBioscience | Cat. #: 47-0081-80; RRID:AB_1272221 | Clone 53-6.7 |
| Antibody | Anti-mouse CD4 BV605 | BioLegend | Cat. #: 100548; RRID:AB_2563054 | Clone RM4-5 |
| Antibody | Anti-mouse CD4 PE-Cy7 | Tonbo Biosciences | Cat. #: 60-0042; RRID:AB_2621829 | Clone RM4-5 |
| Antibody | Anti-mouse CD44 BV786 | BD Biosciences | Cat. #: 563736; RRID:AB_2738395 | Clone IM7 |
| Antibody | Anti-mouse CD44 FITC | eBioscience | Cat. #: 11-0441-85; RRID:AB_465046 | Clone IM7 |
| Antibody | Anti-mouse CD44 rf710 | Tonbo Biosciences | Cat. #:80-0441; RRID:AB_2621985 | Clone IM7 |
| Antibody | Anti-mouse CD45.2 FITC | Tonbo Biosciences | Cat. #: 35-0454; RRID:AB_2621692 | Clone 104 |
| Antibody | Anti-mouse CD45.1 PE-Cy7 | Tonbo Biosciences | Cat. #: 60-0453; RRID:AB_2621850 | Clone A20 |
| Antibody | Anti-mouse CD90.1 ef450 | eBioscience | Cat. #: 48-0900-82; RRID:AB_1272254 | Clone HIS51 |
| Antibody | Anti-mouse CD90.2 PE-Cy7 | Tonbo Biosciences | Cat. #: 60-0903; RRID:AB_2621857 | Clone 30-H12 |
| Antibody | Anti-mouse MHC Class II (I-A/I-E) APC ef780 | Thermo Fisher Scientific | Cat. #: 47-5321-82; RRID:AB_1548783 | Clones M5/114.15.2 |
| Antibody | Anti-mouse MHC Class II (I-A/I-E) BV510 | BioLegend | Cat. #: 107635; RRID:AB_2561397 | Clones M5/114.15.2 |
| Antibody | Anti-mouse F4/80 BV510 | BioLegend | Cat. #: 123135; RRID:AB_2562622 | Clone BM8 |
| Antibody | Anti-mouse F4/80 APC ef780 | Thermo Fisher Scientific | Cat. #: 47-4801-82; RRID:AB_2735036 | Clone BM8 |
| Antibody | Anti-mouse CD122 BV421 | BD Biosciences | Cat. #: 562960; RRID:AB_2737918 | Clone TM-β1 |
| Antibody | Anti-human Granzyme B PE | Invitrogen/Thermo | Cat. #: GRB04; RRID:AB_2536538 | Clone GB11 |
| Antibody | Anti-mouse IFN-g PE-Cy7 | Tonbo Biosciences | Cat. #: 60-7311-U100; RRID:AB_2621871 | Clone XMG1.2 |
| Antibody | Anti-mouse TNFa APC | eBioscience | Cat. #: 17-7321-81; RRID:AB_469507 | Clone MP6-XT22 |

*Continued on next page*

*Continued*

| Reagent type (species) or resource | Designation | Source or reference | Identifiers | Additional information |
|---|---|---|---|---|
| Antibody | Anti-mouse CD107a APC | BioLegend | Cat. #: 121614; RRID:AB_2234505 | Clone ID4B |
| Antibody | Anti-mouse IL-2 APC | BioLegend | Cat. #: 503810; RRID:AB_315304 | Clone JES6-5H4 |
| Antibody | Anti-mouse CD62L BV510 | BD Biosciences | Cat. #: 563117; RRID:AB_2738013 | Clone MEL-14 |
| Antibody | Anti-mouse CD62L PerCP-Cy5.5 | Tonbo Biosciences | Cat. #: 65-0621 | Clone MEL-14 |
| Antibody | Anti-mouse KLRG1 PE-Cy7 | eBioscience/Thermo | Cat. #: 25-5893-82; RRID:AB_1518768 | Clone 2F1 |
| Antibody | Anti-mouse CD127 BV786 | BD Biosciences | Cat. #: 563748; RRID:AB_2738403 | Clone SB/199 |
| Antibody | Anti-mouse CD69 FITC | Tonbo Biosciences | Cat. #: 35-0691; RRID:AB_2621698 | Clone H1.2F3 |
| Antibody | Anti-mouse CD25 PE-Cy7 | Thermo Fisher | Cat. #: 25-0251-82; RRID:AB_469608 | Clone PC61.5 |
| Antibody | Anti-mouse CD103 BV510 | BD Biosciences | Cat. #: 563087; RRID:AB_2721775 | Clone M290 |
| Antibody | Anti-mouse PD-1 PerCP-Cy5.5 | BioLegend | Cat. #: 135208; RRID:AB_2159184 | Clone 29F.1A12 |
| Antibody | Anti-mouse PD-1 FITC | BioLegend | Cat. #: 135214; RRID:AB_10680238 | Clone 29F.1A12 |
| Antibody | Anti-mouse Tim-3 BV421 | BioLegend | Cat. #: 119723; RRID:AB_2616908 | Clone RMT3-23 |
| Antibody | Anti-mouse CTLA-4 PE-Cy7 | BioLegend | Cat. #: 106314; RRID:AB_2564238 | Clone UC10-4B9 |
| Antibody | Anti-mouse CD5 APCef780 | Thermo Fisher scientific | Cat. #: 47-0051-82; RRID:AB_2573940 | Clone 53-7.3 |
| Antibody | Anti-mouse TCRβ APCef780 | Thermo Fisher Scientific | Cat. #: 47-5961-82; RRID:AB_1272173 | Clone H57-597 |
| Antibody | Anti-mouse LAG3 PE-Cy7 | BioLegend | Cat. #: 125226; RRID:AB_2715764 | Clone C9B7W |
| Antibody | Anti-mouse CD49a BUV395 | BD Biosciences | Cat. #: 740262; RRID:AB_2740005 | Clone Ha31/8 |
| Antibody | Anti-mouse CXCR3 PerCP-Cy5.5 | BioLegend | Cat. #: 126514; RRID:AB_1186015 | Clone CXCR3-173 |
| Antibody | InVivoMab anti-mouse IL-2 antibody | BioXCell | Cat. #: BE0043-1; RRID:AB_1107705 | Clone S4B6 |
| Chemical compound, drug | Recombinant Mouse IL-2 Protein, CF | R&D Systems | Cat. #: 402-ML-500/CF | |
| Chemical compound, drug | Streptavidin, R-Phycoerythrin Conjugate, premium grade | Invivogen | Cat. #: S21388 | |
| Chemical compound, drug | Streptavidin, Allophycocyanin Conjugate, premium grade | Invivogen | Cat. #: S32362 | |
| Chemical compound, drug | H-2K(b) SVYDFFVWL Monomer | NIH Tetramer Core | | Biotinylated Monomer |

*Continued on next page*

*Continued*

| Reagent type (species) or resource | Designation | Source or reference | Identifiers | Additional information |
|---|---|---|---|---|
| Chemical compound, drug | Ghost Dye Red 780 Viability Dye | Tonbo Biosciences | Cat. #: 13-0865-T100 | |
| Chemical compound, drug | LIVE/DEAD Fixable Aqua Dead Cell Stain Kit | Thermo Fisher | Cat. #: L34966 | |
| Chemical compound, drug | Annexin V FITC Apoptosis Detection Kit | BD Biosciences | Cat no. 556570; RRID:AB_2869085 | |

## Mice

C57BL/6 (WT) mice were obtained from Charles River laboratories and housed in specific pathogen-free conditions at the University of Minnesota. $Dct^{-/-}$ mice on a C57BL/6 background were developed by Katie Stagliano and A. Andy Hurwitz at the NCI (see below); the mice were subsequently bred in-house on different congenic backgrounds and housed in specific pathogen-free conditions. Animals were used at 6–14 weeks of age. All animal experiments were approved by the Institutional Animal Care and Use Committee at the University of Minnesota. In accordance with NIH guidelines, both male and female animals were used in experiments; males are indicated by square symbols in the figures.

## Generation of $Dct^{\Delta 2\text{-}6}$ ($Dct^{-/-}$) mice

The construct used to generate an exon 2–6 deletion mutant of *Dct* ($Dct^{\Delta 2\text{-}6}$—referred to as $Dct^{-/-}$ in the manuscript) was synthesized by IDT (Coralville, IA). Electroporation of the gene construct into mouse embryonic stem cells (mESCs) was performed by the Transgenic Mouse Model Laboratory (TMML) of the Laboratory Animal Services Program at the National Cancer Institute, Frederick, MD. Polymerase chain reaction (PCR) primers were designed to screen for the presence of the recombined construct in the proper genomic location of prospective clones. pDRAW32 deoxyribonucleic acid (DNA) analysis software (http://www.acalone.com) and Primer3 (http://www.simgene.com) were used to design primers, and the NCBI Primer-BLAST tool was used to confirm the specificity of primers in the mouse genome (https://www.ncbi.nlm.nih.gov/tools/primer-blast/).

Genomic DNA from candidate mESC samples was purified using standard phenol chloroform extraction. Clones were screened by PCR using the primer pairs listed in *Figure 1—source data 2*. Hot-Start Taq Blue Master Mix from Denville Scientific (Holliston, MA) was used with the following conditions: 5 min at 95°C; followed by 40 cycles of 1 min at 95°C, 1 min at 55°C, and 7 min at 72°C; followed by cooling and storage at 4°C. PCR products were visualized by agarose gel electrophoresis. After screening ~350 mESC samples, four candidates that showed recombination were found. These cells were then transferred into blastocysts and inserted into pseudo-pregnant females by the TMML. Ultimately, two chimera lines were generated and transferred into the Hurwitz laboratory's mouse colony where we oversaw the breeding of chimeras and the intercrossing of pups. Pups were screened for slate coat color, which is the phenotype expected for mice with a homozygous deletion of Dct. Tail clips were used to test for the presence or absence of exons by PCR. Primer pair sequences are listed in *Figure 1—source data 3*. Hot-Start Taq Blue Master Mix from Denville Scientific (Holliston, MA) was used with the following conditions: 5 min at 95°C; followed by 30 cycles of 1 min at 95°C, 1 min at 60°C, and 1 min at 72°C; followed by cooling and storage at 4°C. All primers were purchased from Integrated DNA Technologies, Inc (Coralville, IA).

## Tetramer enrichment

Tetramer enrichment was used to isolate antigen-specific cells from pre-immune or acutely challenged mice. A modification of the method used by *Obar et al., 2008* was employed. Following digestion with collagenase D, single-cell suspensions were prepared from the spleens (acutely challenged mice) or spleen and cervical, axillary, brachial, inguinal, and mesenteric lymph nodes (pre-immune mice). When possible, the same tetramer (Trp2$_{180-188}$/K$^b$) was used in both APC and PE to

ensure specificity. Anti-PE and anti-APC beads and magnetized columns (both from Miltenyi Biotec) were used to enrich for tetramer-bound cells. Samples were stained and analyzed by flow cytometry; CountBright counting beads (Invitrogen) were used for enumeration.

### In vivo priming with Trp2

TriVax immunization was used as previously described (*Cho and Celis, 2009*); mice were immunized intravenously (via tail vein injections) with $Trp2_{180-188}$ peptide or Trp2 and $B8R_{20-27}$ peptides, agonist-anti CD40 antibody (BioXCell), and vaccine-grade poly(I:C), a toll-like receptor 3 agonist (InvivoGen). Peptide doses of 50, 100, and 200 µg per mouse were used for effector time points, transfer experiments, and acute time points, respectively, unless otherwise noted. Animals that received TriVax immunization via intraperitoneal instead of intravenous injection were removed from the analysis unless otherwise noted.

### Infections with LmTrp2

Frozen stocks of LmTrp2 (*Bruhn et al., 2005*) were thawed and grown to log-phase in tryptic soy broth supplemented with streptomycin (50 µg/mL). Mice were typically injected with $\sim 10^5 - 10^6$ colony-forming units intravenously or intraperitoneally. Infectious doses were verified by colony counts on tryptic soy broth-streptomycin plates.

### Ex vivo stimulation

In some experiments, splenocytes were stimulated ex vivo after isolation from infected mice. Splenocytes were incubated with Trp2 peptide ($10^{-6}$ M) and Golgiplug (BD Biosciences) for 4–6 hr at 37℃; parallel wells with no peptide were used as a control. Cells were washed and stained with surface antibodies, followed by fixation and permeabilization with a FoxP3 Fix/Perm kit (eBioscience) or FoxP3/transcription factor staining buffer kit (Tonbo Biosciences) and staining with intracellular antibodies.

### Skin harvests

Skin was shaved and dissected from the underlying tissue, then cut into small pieces and digested in a Collagenase Type III solution. Subsequently, samples were processed using a gentleMACS dissociator (Miltenyi Biotec), filtered, and washed before staining for flow cytometry.

### Adoptive transfer experiments

Bulk polyclonal $CD8^+$ T cells were isolated from the spleen and lymph nodes of WT or $Dct^{-/-}$ mice using negative magnetic enrichment ($CD8a^+$ T cell Isolation Kit; Miltenyi Biotec). Enriched $CD8^+$ T cells (typically ~75–90% pure) were resuspended in sterile PBS and $2-2.5 \times 10^6$ $CD8^+$ T cells were injected intravenously per recipient mouse; recipient mice were congenically distinct (by CD45 and/ or Thy-1 alleles). One day later, the recipient mice were immunized with TriVax or LmTrp2 intravenously or intraperitoneally. Mice were sacrificed for analysis 7 days later.

### Bulk RNA sequencing of pre-immune mice

$Trp2/K^b$-specific cells were isolated from pre-immune WT and $Dct^{-/-}$ mice using tetramer enrichment followed by FACS on double tetramer-positive cells. Cells were isolated from three separate cohorts, with each cohort comprising eight WT and eight $Dct^{-/-}$ mice. The Clontech StrandedRNA Pico Mammalian workflow was used for library preparation, and samples were sequenced using an Illumina NextSeq instrument ($2 \times 75$ bp paired end reads).

### Bulk RNAseq analysis

Raw sequencing data were demultiplexed by sample into FASTQs (mean 24.6 million reads/sample) and mapped against the mouse genome (Ensembl GRCm38 release 95) using Hisat2 software (v 2.1.0). Gene level quantification was completed using Subread featureCounts software (v 1.6.2) and the read counts table was processed in R (v 3.5.2). Differentially expressed genes were identified with DESeq2 software (v 1.22.2) using a negative binomial model with effect size estimation completed by apeglm algorithm via the lfcShrink function. Group comparison p-values were adjusted by

the Benjamini and Hochberg method to account for multiple hypothesis testing where genes with a false discovery rate (FDR) q < 0.05 were investigated in downstream analyses.

## Single-cell RNA sequencing

Day 7: WT and $Dct^{-/-}$ mice were primed with TriVax (50 µg Trp2), and Trp2/$K^b$-specific cells were isolated using negative enrichment for $CD8^+$ T cells followed by FACS for tetramer-positive cells. Day 3: WT and $Dct^{-/-}$ mice were primed with TriVax (200 µg Trp2), and Trp2/$K^b$-specific cells were isolated using tetramer enrichment followed by FACS. At both time points, cells were submitted for barcoding and library preparation using the 10× Genomics platform (Chromium Single Cell 5′ Library and Gel Bead Kit; *Zheng et al., 2017*), and samples were sequenced using an Illumina Nova-Seq instrument with 2× 150 bp paired end protocol.

## Single-cell RNAseq analysis

Raw sequencing data were processed using Cell Ranger (v 3.0.2; 10× Genomics) software programs 'mkfastq' for demultiplexing the WT and $Dct^{-/-}$ Illumina libraries and 'count' for read alignment against the mouse genome (mm10, provided by 10× Genomics, v 3.0.0) and generation of the mRNA transcript count table. Raw count data were loaded into R (v 3.6.1) and analyzed with the Seurat R package (v 3.0.3.9039) (*Butler et al., 2018*; *Stuart et al., 2019*). All scRNA datasets (WT or $Dct^{-/-}$) at each time point (3 or 7 days after TriVax immunization) were independently filtered to include only cells (i.e., uniquely barcoded transcripts) expressing more than 300 genes and genes expressed in more than three cells (e.g., counts > 0). The proportion of mitochondrial RNA in each cell was calculated and cells with extreme levels (top or bottom 2% of all cells) were removed from the analysis. Genes with extreme expression levels (top or bottom 1% of all genes) were removed. Contaminating cells in the day 3 dataset expressing high levels of B cell or myeloid lineage marker genes and low levels of T cell markers were removed using empirically derived thresholds (675 B cells and 26 myeloid cells removed from WT and 117 B cells removed from $Dct^{-/-}$). Downstream analysis of the day 3 dataset included a total of 4539 WT cells (19,326 genes) and 11,680 $Dct^{-/-}$ cells (19,416 genes); the day 7 dataset included a total of 6254 WT cells (12,902 genes) and 4784 $Dct^{-/-}$ cells (12,437 genes). The datasets from each time point were analyzed similarly in parallel, unless otherwise noted. Raw RNA counts were normalized with the LogNormalize function and each cell was classified according to its expression of canonical cell cycle genes using the CellCycleScoring function from Seurat (S-phase and G2/M-phase gene sets provided by Seurat were originally developed by *Tirosh et al., 2016*). For each cell, a cell cycle score was computed by subtracting the average expression of a random control gene set from the average expression of either S or G2/M cell cycle gene sets. If the difference score was negative for both comparisons, the cell was labeled G1. Otherwise, the cell was labeled according to which difference score was larger (S or G2/M). Raw RNA counts were normalized and transformed using the Seurat SCTransform function (*Hafemeister and Satija, 2019*) including the percent of mitochondria expression as a regression factor. Principal components analysis (PCA) was performed using the normalized, mean-centered, and scaled SCT dataset (RunPCA function). The top 3000 variable genes from each dataset were identified using the FindVariableFeatures function (vst method) and were used for WT and $Dct^{-/-}$ sample integration (*Stuart et al., 2019*). Two-dimensional projections were generated using the top 30 PCA vectors as input to the RunTSNE and RunUMAP functions. Cells were clustered using the FindNeighbors (top 30 PCA vectors) and FindClusters functions (testing a range of possible resolutions: 0.2, 0.4, 0.8, 1.2, 1.6). Pairwise DE testing (Wilcox rank-sum) with the FindMarkers function was performed between all initial clusters; any two clusters were merged if there were fewer than five significant DE genes (i.e., absolute value of $\log_2$-fold-change≥0.25 and Bonferroni adjusted p-value≤0.01). Pairwise DE testing continued on subsequently merged clusters. A final resolution of 0.2 was chosen (merging of initial clusters by DE testing was not required) to best represent the biological processes within both datasets. Cluster-specific pathway expression testing was completed using the VISION R package (*DeTomaso et al., 2019*) and figures were generated using the ggplot2 R package (*Wickham, 2016*). Gene Set Enrichment Analysis (*Mootha et al., 2003*; *Subramanian et al., 2005*) was performed using pre-ranked gene lists (sorted from largest to smallest $\log_2$-fold change between clusters compared). Gene set enrichment statistics were calculated for two gene set collections in the Molecular Signatures Database (hallmarks and c2 curated) derived

for mouse symbols using the R package msigdbr, v 6.2.1 using the R package clusterProfiler (v 3.12.0). Interesting gene sets with an FDR q < 0.05 were evaluated. Raw and processed data have been deposited at Gene Expression Omnibus and are available via GEO.

## IL2 complex treatment

Ten micrograms antibody (S4B6-1; Bio XCell) plus 1 µg murine recombinant carrier-free murine IL-2 (R and D) was administered per mouse via intraperitoneal injection on day 5 after priming with Tri-Vax or LmTrp2; control mice received an equal volume of PBS.

## Vitiligo induction

Donor mice (WT and $Dct^{-/-}$) were primed with TriVax (100 µg Trp2); in one experiment, donor mice received ~50% less of the other TriVax components to minimize adverse reactions. Negative enrichment for CD8$^+$ T cells was performed on day 7. Live cells were counted using a hemocytometer, and the percentages of CD8$^+$ and Trp2/K$^b$ tetramer-binding cells were applied to enumerate Trp2/K$^b$-specific cells; equal numbers (50,000) of WT or $Dct^{-/-}$ Trp2/K$^b$-specific cells were transferred to WT or P14 recipients. Recipients were treated with TriVax (100 µg Trp2) later the same day. On day 6 after cell transfer, the recipients were bled to assess donor populations. The mice were then treated with DNFB (0.15% in 4:1 acetone:olive oil) on the left flank; 30 µL was applied to a shaved patch of skin ~1.5 cm×1.5 cm in size; 30 µL of vehicle was applied to the right flank in the same manner. Control mice did not receive cell transfers, but did receive TriVax immunization and DNFB treatment at the same time as mice receiving cell transfers. Mice were monitored for vitiligo development on a weekly basis by an observer blinded to the experimental groups or by two independent observers.

## Tetramers and flow cytometry

H-2K$^b$ tetramers loaded with Trp2$_{180-188}$ or B8R$_{20-27}$ were obtained from the NIH tetramer core facility and labeled with streptavidin-fluorophore conjugates in house. Single-cell suspensions were stained with tetramers (when applicable) and fluorescent dye-conjugated antibodies purchased from BD Biosciences, Tonbo Biosciences, eBioscience, or BioLegend. In many experiments, Live/Dead Fixable Aqua Dead Cell Stain Kit (Thermo Fisher Scientific) was used for dead cell exclusion. When applicable, cells were fixed with a FoxP3 Fix/Perm kit (eBioscience) or FoxP3/transcription factor staining buffer kit (Tonbo Biosciences). These kits were also used for permeabilization prior to staining with intracellular antibodies. Samples were run on a BD LSR II or BD Fortessa instrument using BD FACSDiva (BD Bioscience), and data were analyzed with FlowJo (BD).

## Statistical analysis

Initial sample size estimates were based on use of G*Power software (University of Dusseldorf). Prism software (GraphPad) was used to plot data and conduct statistical analyses. An unpaired t test was used for two-way comparisons between two groups. A one-way ANOVA with Sidak's or Tukey's multiple comparisons test was used when multiple comparisons were performed. Log-rank (Mantel-Cox) tests were used to evaluate Kaplan-Meier curves. A two-way ANOVA with Tukey's multiple comparisons test was used to evaluated vitiligo scores over time. p-Values are represented as follows: $*p<0.05$, $**p<0.01$, $***p<0.001$, $****p<0.0001$.

## Acknowledgements

We thank members of the Jamequist lab for helpful discussions and technical assistance. We thank the UMN Flow Cytometry Resource for cell sorting, the UMN Genomics Center for assistance with sequencing experiments, and the NIH tetramer core for peptide/MHC monomers. We thank the Celis and Restifo labs for providing reagents and samples used in enumerating cells specific for alternative melanocyte epitopes. We thank Dietmar Zehn for reviewing an early draft of this manuscript.

This work was supported by NIH grants P01 AI035296 (SCJ), R01 AI140631 (SCJ), and T32 OD010993 (ENT).

# Additional information

## Competing interests

Arthur A Hurwitz: is affiliated with AgenTus Therapeutics, Inc. The author has no financial interests to declare. Ross B Fulton: is affiliated with HiFiBio, Inc. The author has no financial interests to declare. The other authors declare that no competing interests exist.

## Funding

| Funder | Grant reference number | Author |
| --- | --- | --- |
| National Institute of Allergy and Infectious Diseases | R01AI140631 | Stephen C Jameson |
| National Institute of Allergy and Infectious Diseases | P01AI035296 | Stephen C Jameson |
| NIH Office of the Director | T32OD010993 | Emily N Truckenbrod |
| Cancer Research Institute | | Ross B Fulton |
| National Cancer Institute | T32CA009138 | Kristin R Renkema |

The funders had no role in study design, data collection and interpretation, or the decision to submit the work for publication.

## Author contributions

Emily N Truckenbrod, Conceptualization, Formal analysis, Validation, Investigation, Visualization, Methodology, Writing - original draft, Writing - review and editing; Kristina S Burrack, Data curation, Validation, Investigation, Methodology, Writing - review and editing; Todd P Knutson, Data curation, Formal analysis, Validation, Visualization, Writing - review and editing; Henrique Borges da Silva, Formal analysis, Investigation, Writing - review and editing; Katharine E Block, Formal analysis, Validation, Investigation, Writing - review and editing; Stephen D O'Flanagan, Investigation; Katie R Stagliano, Conceptualization, Resources, Investigation, Writing - review and editing; Arthur A Hurwitz, Conceptualization, Resources, Methodology, Writing - review and editing; Ross B Fulton, Resources, Formal analysis, Investigation, Methodology, Writing - review and editing; Kristin R Renkema, Conceptualization, Data curation, Formal analysis, Supervision, Validation, Investigation, Visualization, Methodology, Project administration, Writing - review and editing; Stephen C Jameson, Conceptualization, Resources, Supervision, Funding acquisition, Writing - original draft, Project administration, Writing - review and editing

## Author ORCIDs

Emily N Truckenbrod (iD) https://orcid.org/0000-0002-3819-6307
Todd P Knutson (iD) http://orcid.org/0000-0001-8431-9964
Stephen C Jameson (iD) https://orcid.org/0000-0001-9137-1146

## Ethics

Animal experimentation: This study was performed in strict accordance with the NIH Guide for the Care and Use of Laboratory Animals and handled according to protocols approved but the University of Minnesota IACUC (#1709-35136A and #2007-38243A).

## Decision letter and Author response

Decision letter https://doi.org/10.7554/eLife.65615.sa1
Author response https://doi.org/10.7554/eLife.65615.sa2

## Additional files

### Supplementary files
• Transparent reporting form

### Data availability
NextGen sequencing data has being deposited at GEO: Code GSE171221.

The following dataset was generated:

| Author(s) | Year | Dataset title | Dataset URL | Database and Identifier |
|---|---|---|---|---|
| Truckenbrod EN, Burrack KS, Knutson TP, Block KE, Stagliano KR, Hurwitz AA, Fulton RB, Renkema KR, Jameson SC | 2021 | Non-deletional CD8+ T cell self-tolerance permits responsiveness but limits tissue damage | http://www.ncbi.nlm.nih.gov/geo/query/acc.cgi?acc=GSE171221 | NCBI Gene Expression Omnibus, GSE171221 |

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
