## [Decision Letter]

**Acceptance summary:**

This is a very interesting manuscript which aims to identify distinguishing features of cells that undergo self tolerance. The authors have developed a novel system involving a new T cell receptor transgenic mouse to do this. The work is elegant and indicates that there are few differences between tolerized cells and non-tolerized cells. Interestingly, the authors demonstrate that regardless of tolerization, cells proliferate in response to antigen but tolerized cells prematurely fail to continue to proliferate to the same extent as non-tolerized cells. Most tolerance models are highly contrived and this study represents a significant effort to generate a more physiological system. The authors have provided a detailed and very thorough response to the comments of the reviewers. This work will be a valuable addition to the field.

**Decision letter after peer review:**

Thank you for submitting your article "Non-deletional CD8^+^ T cell self-tolerance permits responsiveness but limits tissue damage" for consideration by *eLife*. Your article has been reviewed by 3 peer reviewers, one of whom is a member of our Board of Reviewing Editors, and the evaluation has been overseen by Satyajit Rath as the Senior Editor. The following individuals involved in review of your submission have agreed to reveal their identity: Emma Hamilton-Williams (Reviewer #2); Ian A Parish (Reviewer #3).

Essential Revisions:

1. A number of sections require more extensive phenotyping in the context of the activation setting and across different tissues.

2. The Dct^-/-^ mouse requires much more extensive description and characterisation as outlined by the reviewers.

3. The proposed mechanism is not fully supported, thus the authors are asked to provide stronger molecular/cellular data to support their claim, and balance other deletional/tolerance possibilities in discussing the outcomes and interpretations of their work.

*Reviewer #1:*

This is a very interesting manuscript which aims to identify distinguishing features of cells that undergo self tolerance. The authors have developed a new and novel system involving a new T cell receptor transgenic mouse to do this. The work is elegant and indicates that there are few differences between tolerized cells and non-tolerized cells. Interesting the authors demonstrate that regardless of tolerization, cells proliferate in response to antigen but tolerized cells prematurely fail to continue to proliferate to the same extent as non-tolerized cells.

1. Both males and females were used in experiments – were differences evaluated between these two sexes? or expected?

2. Were cells stained for Lag3 and CTLA-4 expression?

3. RNAseq was performed but few genes were proffered as regulating the early termination of proliferation. This was performed on day 3, have cells a few days later been analysed or key genes interrogated? Are cytotoxic genes amplified at the same rate in tolerized vs non-tolerized cells?

4. CD25 expression is postulated as a mechanism of regulation. However, CD25 expression is highly variable, even between different infectious settings and thus it is unclear that it might be the defining feature.

5. This work is elegant but seems to lack a strong mechanism explaining the outcomes for tolerized cells and how these might be different from non-tolerized cells.

*Reviewer #2:*

Truckenbrod et al. evaluate tolerance induction in endogenous melanocyte antigen-specific CD8^+^ T cells from mice with and without expression of the specific self-antigen (Trp2). They show that while knock-out mice still have ample antigen-specific cells present (albeit in reduced numbers and with lower tetramer binding) they appeared phenotypically naïve. Upon immunisation, the antigen-specific cell population still expands dramatically (1000x) but with slower proliferation kinetics and the proliferation could be recovered by providing IL-2. Thy use single cell sequencing to identify that the most highly proliferative cell clusters are greatly reduced in the WT mice. They use a disease model of vitiligo induction and show that the cells from WT mice with endogenous antigen exposure are less able to induce disease. The strengths of the paper are a very thorough phenotypic analysis including using sophisticated single cell technology to elucidate the responding populations. They also validated their findings using both a vaccination model and an infection model. The model is highly relevant to human disease, particularly in the setting of cancer immunotherapy when self-antigen specific T cell populations must be recruited to the anti-tumour response. The paper is well written and clearly presented. The weakness was the inability to identify phenotypic differences in the pre-immune T cell response that programmed the difference in tolerance outcomes, which did not appear to be derived from recruitment of different affinity cells into the response (though this was not directly tested and closer analysis of TCR usage or clonotype sequencing may reveal this as a potential mechanism). A lack of gene expression differences in the pre-immune population may have been due to analysis of combined cell populations from multiple immune sites (spleen, peripheral and mesenteric LN). While presumably, antigen experience was occurring in the skin-draining LN where Trp2 antigen would be present. This may have diluted their ability to observe phenotypic differences/signs of antigen experience in the pre-immune population. Antigen exposure in the periphery may have resulted in reprogramming of the signalling machinery rather than gene expression changes. However, the authors do not claim to have determined the mechanism behind the tolerogenic phenotype and will investigate this in future studies. overall, the findings are interesting and novel.

My key issue was combining analysis of spleen and various LN in the analysis of the pre-immune population. Combining these different cell sources may well have masked the ability to detect a population with signs of antigen experience. Phenotyping of activation markers in skin-draining LN compared to other LN/spleen where Trp2 is not expected to drain should be included.

I am not sure that the conclusion from Figure 4D/E that there is no preferential response of higher affinity cells in the KO compared to the WT can be made from this data. It is hard to compare MFI between the two genotypes with low numbers of mice when small differences in staining efficiency can amplify differences between the samples. Ideally, the staining here needs to be normalised to an internal control within the same sample (total TCR staining?) or the two genotypes should be mixed and stained together.

Figure 1B and D are the histograms gated on tetramer+ cells? It is hard to compare Figure 1B and Suppl. Figure 1A as tick marks are not clear/no numbers on axis. In Figure 1B and 1C are male KO vs female WT mice compared or does the legend comment that squares represent male mice only apply to panel A and D?

Figure 2E: is this tetramer staining from enriched or non-enriched populations?

Methods: please explicitly state which lymph nodes are included in 'peripheral' lymph nodes.

Methods: please add a description of how the cell-cycle stage is assessed from the single-cell data.

Methods: Please provide details of the congenic markers used in the mice.

Suppl Figure 1A: please include example of WT vs KO pre-immune tetramer staining.

*Reviewer #3:*

Truckenbrod et al. examine how immune tolerance processes impact upon an endogenous population of self-reactive CD8^+^ T cells specific for the Trp2 self-peptide. They demonstrate persistence of a population of self-reactive CD8^+^ T cells in wild-type mice, albeit at lower numbers and avidity compared to those cells found within a mouse lacking the target self-antigen. This self-reactive T cell population could be expanded upon antigen encounter during either vaccination or infection, although interestingly the expanded T cell populations that developed in wild-type mice bearing the self-antigen exhibited some defects early during activation, and were less capable of causing autoimmune disease, compared to cells from mice lacking the target self-antigen. These defects were cell intrinsic and apparently not due to suppression by antigen-specific regulatory T cells. The authors conclude that a novel tolerance mechanism independent of deletion preserves responsiveness of cells to antigenic challenge but restrains their capacity to cause tissue damage.

Strengths

The key strength of this manuscript is the experimental model used. They have examined tolerance within endogenous T cell populations specific for a natural self-antigen in wild-type mice, and in parallel they examine control mice that lack the self-antigen to enable quantification of tolerance. This study is important as most tolerance phenomena have been described in far more artificial settings (eg. using high affinity TCR transgenic cells and/or studying tolerance to model self-antigens). By examining tolerance in more physiological models, the real-world relevance of phenomena previously described in more artificial tolerance models can be assessed. Their data suggest that tolerance within endogenous repertoires is more complex than that observed in more artificial models, consistent with recent findings in humans, and are thus a valuable addition to the field.

The authors have also conducted a comprehensive and impressive set of experiments to thoroughly probe the effect of tolerance on immune responsiveness in their models.

Weaknesses

The main weakness of this manuscript is that the argument that the observed phenomena cannot be explained by previously described tolerance mechanisms such as deletion and ignorance are not supported by the data. They argue that antigen exposure potentially leads to epigenetic conditioning of the remaining cells such that they can respond to antigenic challenge but cannot cause disease. However, their data do provide strong evidence that high avidity self-reactive cells are deleted by tolerance mechanisms and that the remaining cells are simply lower avidity cells that fall below the threshold for selection. These residual cells appear naïve, likely because they receive insufficient signals to be activated (ie. they are ignorant). This shift towards a lower avidity population is just as likely an explanation for the observed phenotypes, particularly given that high avidity cells are likely to be more destructive in autoimmune models. Thus, the findings can be explained by both deletion and ignorance, which are previously reported phenomena.

The authors also study the autoimmune potential of tolerized or control cells in a vitiligo model by first expanding the cells by vaccination prior to transfer into the disease model. However, little information is provided on the phenotype of these expanded cells prior to transfer. This is important as the phenotypes are likely to be different given that there is an expansion defect in tolerized cells. Phenotypic differences at this stage could explain differences in autoimmune potential. Additionally, the authors have not examined the skin infiltrating potential of the transferred cells, which may also be different due to differences in initial priming.

Finally, the authors have not examined whether there is evidence of migration of high avidity self-reactive cells into the skin in mice that bear the self-antigen vs those that do not. Differences in skin migration could explain the lower cell counts of tolerized cells in the spleen either in the steady-state or after vaccination.

The following changes are required to the manuscript

i. The manuscript should be revised to substantially temper the conclusions. There is clearly some deletion occurring (~2 fold) so it is misleading to suggest that no deletional tolerance has occurred. Given that this reduction in numbers is associated with lower avidity, this argues that the high avidity cells are deleted in WT mice. This is reflected in other data in the paper (eg. appearance of an IRF4hi scRNAseq cluster in KO mice suggestive of stronger TCR signal strength after vaccination, and enhanced expansion of KO cells in WT mice where there is presumably less competition from high avidity cells). The remaining lower avidity cells appear naïve by all measures, and the simplest explanation here is ignorance. While it is possible that other factors may be at play (eg. epigenetic conditioning), without identifying a shared endogenous TCR present in WT and KO mice and showing that it behaves differentially in both contexts (eg. via retrogenic mice), this is very hard to prove. Overall, there should be a diminished focus on an unknown tolerance mechanism contributing to the phenotypes – this can be replaced by a section in the Discussion considering the relative likelihood of deletion/ignorance vs epigenetic conditioning as an explanation for the findings. "Non-deletional" should be removed from the title, and the potential for deletion of high avidity clones to explain the phenotype should not be so readily dismissed within the manuscript.

ii. Do cells end up in the skin in WT mice (ie. the site of Ag)? This hasn't been ruled out as an explanation for reduced cells in the spleen (either after vaccination or in the steady-state). There could be subclinical inflammation/autoimmunity drawing high avidity cells into this site.

iii. More detail on the Dct KO strain is needed (schematic showing region deleted, more explanation on generation, whether or not a truncated protein is produced etc)

iv. Basic phenotyping of d7/8 cells post vaccination is needed, particularly as this has important implications for the vitiligo experiment. Eg. %KLRG1+ vs CD127+KLRG1-, GzmB expression, chemokine receptor expression (CXCR3, CX3CR1), cytokine polyfunctionality (IFNγ/TNFa/IL-2 co-production), degranulation etc. It's likely that the cells transferred in the vitiligo experiments are in different states of effector differentiation given the expansion and avidity differences, which is a major caveat in this experiment that is not considered/discussed. Also, Dct^-/-^ CD8s have less competition from endogenous high avidity Trp2-responsive cells in WT mice, so will likely fare better in this transfer system based on Figure 3. Although there was no significant difference in persistence/expansion at d6, they may persist better long-term in this model, and this point should at least be discussed as an explanation for increased disease.

v. Was there a difference in early skin infiltration of Dct^-/-^ derived CD8s in the vitiligo experiments? And did only higher avidity cells make it into the skin? Differences in chemokine receptor/integrin expression downstream of altered differentiation could affect homing capacity.

vi. Re avidity – although there is no change in relative avidity between the groups, only cells above a certain avidity threshold (~1000 MFI) appear to expand in both groups (compare Figure 2D to 1B). This would suggest that higher avidity cells are preferentially selected for expansion, meaning the statement that there is no selection for high avidity cells is technically incorrect and should be qualified.

vii. Figure 1 – FR4 and CD73 should be included for completeness given that they are bona fide anergy markers. Compiled MFIs of all markers in this figure should also be plotted. Additionally, CD25 MFIs in Figure 4B should be plotted.

---

## [Author Response]

Essential Revisions:1. A number of sections require more extensive phenotyping in the context of the activation setting and across different tissues.2. The Dct^-/-^ mouse requires much more extensive description and characterisation as outlined by the reviewers.3. The proposed mechanism is not fully supported, thus the authors are asked to provide stronger molecular/cellular data to support their claim, and balance other deletional/tolerance possibilities in discussing the outcomes and interpretations of their work.

**We have responded to all of these points in the specific comments below. In summary: (1) we present additional phenotypic (and gene expression) data to extend our analysis; (2) an extensive description of the generation and characterization of the *Dct^-/-^* mice is now included; (3) we expand the discussion of potential mechanisms that underpin tolerance in this system throughout the manuscript, particularly in the discussion.**

Reviewer #1:This is a very interesting manuscript which aims to identify distinguishing features of cells that undergo self tolerance. The authors have developed a new and novel system involving a new T cell receptor transgenic mouse to do this. The work is elegant and indicates that there are few differences between tolerized cells and non-tolerized cells. Interesting the authors demonstrate that regardless of tolerization, cells proliferate in response to antigen but tolerized cells prematurely fail to continue to proliferate to the same extent as non-tolerized cells.

**We thank the reviewer for their enthusiastic comments. Just to clarify, we did not create a new TCR transgenic model in this work (which is focused strictly on responses of the natural polyclonal T cell pool), but we did generate and analyze a novel Dct knockout strain in these studies.**

1. Both males and females were used in experiments – were differences evaluated between these two sexes? or expected?

**We did not anticipate differences between males and females in these studies – we used both in order to comply with NIH guidelines and because susceptibility to some autoimmune diseases does indeed vary between the sexes. We did not conduct stringent statistical analysis comparing responses in males and females but did not observe striking differences. We can withdraw the indication of animal sex from the data if this is considered to be confusing to the readers.**

2. Were cells stained for Lag3 and CTLA-4 expression?

**CTLA-4 and LAG3 staining was shown for pre-immune cells (Figure 1F) and new data show gene expression for these and other exhaustion/anergy markers in primed Trp2/K^b^-specific cells from *Dct^-/-^* and WT mice (Figures S7B, S8C) – in none of these situations did we see substantially increased expression in WT (tolerant) cells, which would have been anticipated if these cells were exhausted.**

3. RNAseq was performed but few genes were proffered as regulating the early termination of proliferation. This was performed on day 3, have cells a few days later been analysed or key genes interrogated? Are cytotoxic genes amplified at the same rate in tolerized vs non-tolerized cells?

**We apologize for the paucity of information on the phenotype and gene expression profiles of Trp2/K^b^specific cells at day 7 post-priming of WT and *Dct^-/-^*, and this was also requested by Rev. 3. As now shown in Figures S3 and S7, we observe only moderate differences between WT and *Dct^-/-^* populations at day 7 (fewer than we observed at day 3); by this stage, which represents the approximate peak of clonal expansion, it is expected that many cells will be out of cycle. As discussed in the manuscript, we expect that reduced early expression of CD25, limiting responsiveness to IL-2, contributes to the reduced expansion of Trp2/K^b^-specific cells in WT mice. Regarding cytotoxicity, we did not see differential expression of the gene encoding Perforin but there was modestly higher expression of transcripts for Granzyme B in KO cells (at day 7 post-priming; Figure S7B and data not shown). We have expanded our description of phenotypic and gene expression characteristics in the text.**

4. CD25 expression is postulated as a mechanism of regulation. However, CD25 expression is highly variable, even between different infectious settings and thus it is unclear that it might be the defining feature.

**This is a fair point – indeed, while CD25 expression contributes to the expansion of CD8^+^ T cells in response to pathogens, this role is fairly limited. However, reduced expression of CD25 has also been noted in other studies on T cell tolerance, including the responses of low affinity/avidity CD8^+^ T cells, and we did observe that IL-2 therapy enhanced the expansion of Trp2/K^b^-specific CD8^+^ T cells in WT mice (and also in *Dct^-/-^* mice)—both in the context of TriVax and LmTrp2 infection.**

5. This work is elegant but seems to lack a strong mechanism explaining the outcomes for tolerized cells and how these might be different from non-tolerized cells.

**We appreciate the reviewer’s comment (shared by Rev. 3). As made clearer in the revised manuscript, our chief goal in this report was to employ a mouse model to investigate whether non-deletional mechanisms dominate in CD8^+^ T cell tolerance toward self-antigens, as was suggested by some studies in humans (Maeda et al. 2014; Yu et al. 2015). By careful analysis in a well-controlled system, we conclude that there is quite modest “pruning” of CD8^+^ T cells with the highest affinity/avidity TCRs for Trp2/K^b^ in WT mice, and that this curtails but does not prevent considerable expansion in response to priming yet has a more substantial effect on restraining induction of immunopathology (the latter being something that could not be tested directly in the human studies). This general conclusion aligns with prior studies on the responses of low affinity/avidity TCR CD8^+^ T cells, but our data suggest the differences in TCR repertoire are remarkably subtle.**

Reviewer #2:[…] The weakness was the inability to identify phenotypic differences in the pre-immune T cell response that programmed the difference in tolerance outcomes, which did not appear to be derived from recruitment of different affinity cells into the response (though this was not directly tested and closer analysis of TCR usage or clonotype sequencing may reveal this as a potential mechanism). A lack of gene expression differences in the pre-immune population may have been due to analysis of combined cell populations from multiple immune sites (spleen, peripheral and mesenteric LN). While presumably, antigen experience was occurring in the skin-draining LN where Trp2 antigen would be present. This may have diluted their ability to observe phenotypic differences/signs of antigen experience in the pre-immune population. Antigen exposure in the periphery may have resulted in reprogramming of the signalling machinery rather than gene expression changes. However, the authors do not claim to have determined the mechanism behind the tolerogenic phenotype and will investigate this in future studies. overall, the findings are interesting and novel.

**As the reviewer mentions, our studies on gene expression or phenotype in the pre-immune pool did not detect substantial differences between Trp2/K^b^-specific cells from WT and *Dct^-/-^* mice. As the reviewer raised the idea that this could be affected by analysis of cells that were combined from multiple lymphoid tissues, we now provide data on this point for the reviewers. This is discussed in detail in the response to specific concerns, but preliminary analysis of skin-draining (vs non-skin-draining) lymph nodes showed similar numbers of Trp2/K^b^-specific cells with similar phenotypic characteristics in WT and *Dct^-/-^* mice.**

**As the reviewer mentions, we do not claim to have defined the mechanism for tolerance in this system – although apparently subtle elimination of T cells with high-affinity/avidity TCRs remains a strong possibility – but we agree that these studies, focusing on the natural polyclonal T cell repertoire and testing the limits of tolerance through to immunopathology, are novel and interesting.**

My key issue was combining analysis of spleen and various LN in the analysis of the pre-immune population. Combining these different cell sources may well have masked the ability to detect a population with signs of antigen experience. Phenotyping of activation markers in skin-draining LN compared to other LN/spleen where Trp2 is not expected to drain should be included.

This is an interesting idea, and we thank the reviewer for raising it. Since we are tracking the natural polyclonal T cell pool in these studies there is, however, a substantial technical barrier in analysis of skin-draining lymph nodes since very few cells would be expected to be Trp2/K^b^ specific. Still, as the reviewer indicated, that assumption might be incorrect – if there were, for example, accumulation of and/or prior activation of Trp2/K^b^-specific CD8^+^ T cells in skin-draining lymph nodes from WT (but not *Dct^-/-^*) mice due to constitutive Trp2 antigen presentation. We have explored this in pilot experiments presented to the reviewers: by pooling skin-draining lymph nodes (axillary, brachial, cervical and inguinal) from 6 mice and performing Trp2/K^b^tetramer enrichment, we could identify sufficient numbers of Trp2/K^b^-specific cells to assess whether there were substantial differences between the populations in WT and *Dct^-/-^* mice. From the same mice, we pooled non-skin draining lymph nodes (mesenteric) separately and also examined splenic cells. We observed that the numbers and cell surface phenotype of Trp2/K^b^tetramer-binding CD8^+^ T cells from skin-draining vs non-skin draining lymph nodes, and from WT vs *Dct^-/-^* mice, were all quite similar: in particular, there was scant evidence for enrichment or prior activation of Trp2/K^b^-specific CD8^+^ T cells isolated in WT skin-draining nodes (preliminary data, not shown). Even with pooling samples from 6 mice, however, we detected only several hundred Trp2/K^b^-specific events per group of lymph nodes (as would be predicted based on these cells being ~0.01% of the preimmune CD8^+^ T cell repertoire), making it hazardous to be dogmatic about these findings. As a result, we would like to limit our description of these preliminary experiments to “data not shown”.

**In addition to these direct studies, it is also worth noting that if it were indeed the case that Trp2/K^b^-specific cells in skin-draining nodes (but not other lymphoid sites) of WT mice were tolerized, this would not account for the poor response of WT CD8^+^ T cells after adoptive transfer into *Dct^-/-^* host mice and Trp2 immunization (Figure 3) – presumably, this population of donor T cells would be overwhelmingly non-tolerant Trp2/K^b^-specific cells from nonskin draining sites. The same set of experiments preclude the idea that a few tolerant Trp2/K^b^-specific CD8^+^ T cells dominantly impair responses (note the robust response of *Dct^-/-^* donor CD8^+^ T cells in WT host mice).**

I am not sure that the conclusion from Figure 4D/E that there is no preferential response of higher affinity cells in the KO compared to the WT can be made from this data. It is hard to compare MFI between the two genotypes with low numbers of mice when small differences in staining efficiency can amplify differences between the samples. Ideally, the staining here needs to be normalised to an internal control within the same sample (total TCR staining?) or the two genotypes should be mixed and stained together.

First, it is important to note that the tetramer MFI ratios do not reflect low numbers of mice – each data point in Figure D is a summary of the average tetramer MFI ratio for groups of mice. So, for example, the d7 timepoint is reflective of a total of 44 mice, and the total for figure 4D is > 150. Second, while the reviewer’s comment is well taken, we showed significant differences in Trp2/K^b^tetramer MFI between WT and *Dct^-/-^* mice in pre-immune mice and all stages of priming in dozens of experiments yet saw no significant difference in the MFIs for the B8R/K^b^-specific cells (a group which happens to be at a similar precursor frequency as Trp2/K^b^ cells but is not likely to be affected by ablation of *Dct*; see Figures 1 and S6). Lastly, we have performed limited assessment of TCRß surface expression in parallel with Trp2/K^b^tetramer binding – as shown in Figure S3A (and preliminary data, not shown), we saw similar TCR expression levels but distinct Trp2/K^b^tetramer staining on the WT vs *Dct^-/-^* cells.

Figure 1B and D are the histograms gated on tetramer+ cells? It is hard to compare Figure 1B and Suppl. Figure 1A as tick marks are not clear/no numbers on axis. In Figure 1B and 1C are male KO vs female WT mice compared or does the legend comment that squares represent male mice only apply to panel A and D?

**Yes, the histograms in 1B and 1D were gated on tetramer-positive cells. We have revised the data shown in Figures 1Band S2A (formerly S1A) to improve comparison.**

Figure 2E: is this tetramer staining from enriched or non-enriched populations?

**Non-enriched. This is now stated in the legend.**

Methods: please explicitly state which lymph nodes are included in 'peripheral' lymph nodes.Methods: please add a description of how the cell-cycle stage is assessed from the single-cell data.Methods: Please provide details of the congenic markers used in the mice.

**As requested we have stated which lymph nodes were harvested (p. 24), how cell-cycle stage was assessed in the scRNAseq studies (p. 26), and the details of the congenic markers used (CD45 and Thy-1 alleles) (p. 25) in the manuscript.**

Suppl Figure 1A: please include example of WT vs KO pre-immune tetramer staining.

**This has been added to Figure S2A (formerly S1A)**

Reviewer #3:[…] The main weakness of this manuscript is that the argument that the observed phenomena cannot be explained by previously described tolerance mechanisms such as deletion and ignorance are not supported by the data. They argue that antigen exposure potentially leads to epigenetic conditioning of the remaining cells such that they can respond to antigenic challenge but cannot cause disease. However, their data do provide strong evidence that high avidity self-reactive cells are deleted by tolerance mechanisms and that the remaining cells are simply lower avidity cells that fall below the threshold for selection. These residual cells appear naïve, likely because they receive insufficient signals to be activated (ie. they are ignorant). This shift towards a lower avidity population is just as likely an explanation for the observed phenotypes, particularly given that high avidity cells are likely to be more destructive in autoimmune models. Thus, the findings can be explained by both deletion and ignorance, which are previously reported phenomena.

**The reviewer raises several important issues. To clarify, we did not intend to convey that we believe epigenetic regulation is the root cause of tolerance in this system, but simply raised this a possibility among several. We would also note that our definition of “ignorance” appears to differ from that of the reviewer – our usage of the term is based on the concept of T cells that are fully competent to respond to an antigen yet do not encounter it during normal homeostasis. This is similar, for example, to the model described by Ohashi and colleagues in 1991 (Ohashi et al., 1991), in which TCR transgenic CD8^+^ T cells maturing in an animal that also expresses the cognate antigen are neither deleted nor impaired in their response to priming with the same antigen. This is distinct from a case in which the T cells are *unable* to respond to the antigen (due, for example, to their TCRs being too low affinity to be activated). We have included a clarification about how we define the term on p. 7 (starting on line 187).**

**Finally, we agree with the reviewer that our data indicate some degree of deletion occurs in WT mice, culling T cells with the highest avidity TCRs for Trp2/K^b^ from the repertoire. While this has certainly been observed before, we believe the novelty and impact of our work is to show that this pruning of the TCR repertoire is quite subtle, leading to a largely overlapping peptide/MHC tetramer staining on tolerant and non-tolerant populations which does not become exaggerated during the immune response. Accordingly, our data highlight that what appears to be relatively modest editing of the polyclonal TCR repertoire can have substantial effects on the capacity of these cells to mediate immunopathology, which we believe to be novel.**

The authors also study the autoimmune potential of tolerized or control cells in a vitiligo model by first expanding the cells by vaccination prior to transfer into the disease model. However, little information is provided on the phenotype of these expanded cells prior to transfer. This is important as the phenotypes are likely to be different given that there is an expansion defect in tolerized cells. Phenotypic differences at this stage could explain differences in autoimmune potential. Additionally, the authors have not examined the skin infiltrating potential of the transferred cells, which may also be different due to differences in initial priming.

**We apologize for not including more data on the Trp2/K^b^-primed populations used for adoptive transfer in the vitiligo studies. We now include phenotypic and gene expression analysis of this population (Figures S3 and S7), which suggest the differences observed between WT and *Dct^-/-^* populations in the early phase of the response to priming are largely resolved by the peak of the immune response. We also have examined infiltration of donor cells into the skin (see next point).**

Finally, the authors have not examined whether there is evidence of migration of high avidity self-reactive cells into the skin in mice that bear the self-antigen vs those that do not. Differences in skin migration could explain the lower cell counts of tolerized cells in the spleen either in the steady-state or after vaccination.

**These are also important points, and we thank the reviewer for raising them. We have now examined the recruitment of donor cells into the skin after TriVax immunization, and these data indicate similar infiltration of WT and *Dct^-/-^* cells in the skin (Figure S4). Additionally, we have added data regarding the number and phenotype of transferred WT and *Dct^-/-^* cells infiltrating the skin in our vitiligo transfer model, which we also find to be comparable regardless of donor origin (Figure S10).**

**Regarding infiltration at steady state, we and others observe extremely low numbers of CD8^+^ T cells in the skin in unmanipulated B6 mice, making it difficult to confidently assess whether any of these are specific for Trp2/K^b^. Likewise, as we outlined in the response to Rev. 2, we do not see convincing evidence for activation of the (rare) Trp2/K^b^ specific CD8^+^ T cells found in the skin-draining lymph nodes of B6 mice at steady state.**

The following changes are required to the manuscripti. The manuscript should be revised to substantially temper the conclusions. There is clearly some deletion occurring (~2 fold) so it is misleading to suggest that no deletional tolerance has occurred. Given that this reduction in numbers is associated with lower avidity, this argues that the high avidity cells are deleted in WT mice. This is reflected in other data in the paper (eg. appearance of an IRF4hi scRNAseq cluster in KO mice suggestive of stronger TCR signal strength after vaccination, and enhanced expansion of KO cells in WT mice where there is presumably less competition from high avidity cells). The remaining lower avidity cells appear naïve by all measures, and the simplest explanation here is ignorance. While it is possible that other factors may be at play (eg. epigenetic conditioning), without identifying a shared endogenous TCR present in WT and KO mice and showing that it behaves differentially in both contexts (eg. via retrogenic mice), this is very hard to prove. Overall, there should be a diminished focus on an unknown tolerance mechanism contributing to the phenotypes – this can be replaced by a section in the Discussion considering the relative likelihood of deletion/ignorance vs epigenetic conditioning as an explanation for the findings. "Non-deletional" should be removed from the title, and the potential for deletion of high avidity clones to explain the phenotype should not be so readily dismissed within the manuscript.

**As mentioned in the previous section (public review), we agree with the reviewer that some measure of clonal deletion (which leads to a reduction of the average number of cells by ~1.4-fold in the pre-immune pool) likely occurs in WT mice, and that the reduced Trp2/K^b^tetramer MFI on these cells as well as other characteristics of their responses (impaired expression of CD25 and IRF4) align with this population being of lower avidity. Indeed, we had discussed our findings in the context of WT cells behaving like low affinity/avidity cells in the original discussion. We do not consider this to be a form of “ignorance” – at least with our intended meaning for the term, which we have clarified in the text. We have also removed the term “non-deletional” from the title and edited the rest of the manuscript to be consistent with our finding that some deletion does occur in WT mice.**

**However, we think it equally important to highlight that the degree of deletion or “pruning” of the TCR repertoire, if this is indeed the basis for tolerance, is remarkably subtle. Although we observe significant differences in the average Trp2/K^b^-specific precursor number and tetramer MFIs, there is considerable overlap between the Trp2/K^b^-specific populations from WT and *Dct^-/-^* mice that would severely limit the ability to predict a given animal’s tolerance status. Hence, our data indicate that assessment of clonal deletion would be an unreliable way to identify self-tolerance in, for example, a clinical setting. These points are made in the discussion.**

**Finally, a central goal of our studies was to model findings that had been published from studies in humans, which suggested minimal clonal deletion among self-specific CD8^+^ T cells and proposed varied mechanistic explanations for tolerance (ranging from a form of anergy to a key role for Tregs) (Maeda et al. 2014; Yu et al. 2015). Our studies indicate that modest editing of the TCR repertoire may minimally impact the magnitude of the response against self-antigen immunization – yet can lead to a more substantial effect on ensuing immunopathology.**

ii. Do cells end up in the skin in WT mice (ie. the site of Ag)? This hasn't been ruled out as an explanation for reduced cells in the spleen (either after vaccination or in the steady-state). There could be subclinical inflammation/autoimmunity drawing high avidity cells into this site.

**As mentioned under public comments, we have examined the recruitment of cells into the skin after immunization (Figures S4 and S10); the number of CD8^+^ T cells we can recover from the skin in unimmunized mice is too low for accurate analysis. We believe the adoptive transfer studies (Figure 3) argue against recruitment to the site of antigen as a basis for the low numbers of Trp2/K^b^-specific splenocytes observed in WT mice – in those studies, WT donor cells expanded to a similar extent in the spleens of mice that did (WT hosts) and did not (*Dct^-/-^* hosts) express the antigen in the skin. The expansion of *Dct^-/-^* donor cells in the spleens of WT host mice was in fact increased relative to *Dct^-/-^* hosts, which is inconsistent with recruitment to the skin depleting the splenic pool of responder cells.**

iii. More detail on the Dct KO strain is needed (schematic showing region deleted, more explanation on generation, whether or not a truncated protein is produced etc)

**We apologize that these details were not provided – a schematic is now shown (Figure S1A), outlining the deleted regions, and details about the generation of the knockout are provided in the methods (p. 23). This knockout allele removes 5 of the 8 coding exons, so we suspect no truncated protein is produced. In any case, the coding sequence for the K^b^-restricted Trp2 epitope (Trp2 180-188) is in the deleted exon 2.**

iv. Basic phenotyping of d7/8 cells post vaccination is needed, particularly as this has important implications for the vitiligo experiment. Eg. %KLRG1+ vs CD127+KLRG1-, GzmB expression, chemokine receptor expression (CXCR3, CX3CR1), cytokine polyfunctionality (IFNγ/TNFa/IL-2 co-production), degranulation etc. It's likely that the cells transferred in the vitiligo experiments are in different states of effector differentiation given the expansion and avidity differences, which is a major caveat in this experiment that is not considered/discussed. Also, Dct^-/-^ CD8s have less competition from endogenous high avidity Trp2-responsive cells in WT mice, so will likely fare better in this transfer system based on Figure 3. Although there was no significant difference in persistence/expansion at d6, they may persist better long-term in this model, and this point should at least be discussed as an explanation for increased disease.

**As mentioned in the public comments, we apologize for the absence of these data. We examined d7 primed WT and *Dct^-/-^* cells by both phenotypic assays and by scRNAseq (Figures S3 and S7). Since the latter provides a more comprehensive assessment of the status of the cells (and also alignment of subpopulations via clustering), we focus on that in the revised manuscript. However, we also show and refer to data regarding most of the phenotypic and functional characteristics mentioned. Overall, and perhaps surprisingly, these data show that by day 7 of TriVax priming the Trp2/K^b^-specific populations in WT and *Dct^-/-^* mice are more closely related than they were at day 3 of priming – the scRNAseq data indicates less distinct clusters (although there certainly are still some minor differences in clustering). This conclusion is amplified by phenotyping – we did not see clear differences in the expression of markers associated with activation or anergy/exhaustion between the populations at this time point. In terms of function, we did observe slightly higher Granzyme B gene expression in *Dct^-/-^* cells, and we saw a slight but significant increase in polyfunctionality in *Dct^-/-^* vs WT cells in ex vivo stimulation assays (in this case, polyfunctional cells made IFN-g and TNFα – we did not see clear production of IL-2, consistent with effector stage cells). The use of KLRG1 as a marker to distinguish short-lived effector- versus memory precursors is difficult in priming with TriVax; as has been published previously, this type of adjuvanted subunit vaccine does not generate a KLRG1^hi^ population (Klarquist et al., 2018). We hope these additional data provide a more comprehensive view of the WT and *Dct^-/-^* cell populations in the late effector phase (these same cells being used for adoptive transfer in the vitiligo studies).**

v. Was there a difference in early skin infiltration of Dct^-/-^ derived CD8s in the vitiligo experiments? And did only higher avidity cells make it into the skin? Differences in chemokine receptor/integrin expression downstream of altered differentiation could affect homing capacity.

**These are interesting but technically challenging experiments. We have had success in isolating donor cells from the skin (both the original DNFB-treated and contralateral flanks) from various timepoints in the vitiligo experiments, and we observe overall similar representation of donor WT and *Dct^-/-^* Trp2/K^b^-specific cells (in terms of both percentage of all infiltrating CD8^+^ T cells and in terms of numbers). These data suggest that it is not infiltration into the skin per se that limits vitiligo induction by WT donor cells. Studies on the tetramer binding of these cells suggested that recruited cells were modestly increased for tetramer staining relative to splenic cells, but this was similar in WT and *Dct^-/-^* donor populations (and directly comparing tetramer staining MFIs on cells extracted from the skin versus those isolated from lymphoid tissues may be difficult). We included these data in Figure S10.**

vi. Re avidity – although there is no change in relative avidity between the groups, only cells above a certain avidity threshold (~1000 MFI) appear to expand in both groups (compare Figure 2D to 1B). This would suggest that higher avidity cells are preferentially selected for expansion, meaning the statement that there is no selection for high avidity cells is technically incorrect and should be qualified.

**We take the reviewer’s point – there is indeed a trend toward increased tetramer staining (often used as a surrogate for TCR avidity) in the Trp2/K^b^-specific cells during the primary response relative to naïve cells, although it is difficult to know if this is truly a selection for a high-avidity population or relates to the activation state of the cells influencing staining. Regardless, the point of our statement was that there is not a widening gap in terms of the relative tetramer MFIs on *Dct^-/-^* and WT cells during the response – i.e., we do not see an *imbalanced* outgrowth of presumed high avidity cells in *Dct^-/-^* relative to WT animals. We have clarified this in the discussion.**

vii. Figure 1 – FR4 and CD73 should be included for completeness given that they are bona fide anergy markers. Compiled MFIs of all markers in this figure should also be plotted. Additionally, CD25 MFIs in Figure 4B should be plotted.

**In our RNA-seq analysis of preimmune cells, FR4 and CD73 were not differentially expressed between Trp2/K^b^-specific cells from WT and *Dct^-/-^* mice. Among Trp2/K^b^-specific cells from immunized mice, very few (< 1.5%) expressed these markers in limited examination by flow cytometry at day 7 after TriVax (data not shown). Small populations of WT Trp2/K^b^ specific cells (< 25% of WT cells in clusters 2 and 3) do express *Izumo1r* (FR4) at day 3 after TriVax by scRNA-seq, but neither marker is differentially expressed in a statistically significant manner at either day 3 or day 7 by scRNA-seq (Figures S7, S8, and data not shown).**

**We have now plotted compiled MFIs for the markers indicated in Figure 1 (Figure S2D) and CD25 MFIs (Figure 4C).**

**References:**

Klarquist, J., Chitrakar, A., Pennock, N. D., Kilgore, A. M., Blain, T., Zheng, C.,... Kedl, R. M. (2018). Clonal expansion of vaccine-elicited T cells is independent of aerobic glycolysis. Sci Immunol, 3(27). doi:10.1126/sciimmunol.aas9822

Ohashi, P. S., Oehen, S., Buerki, K., Pircher, H., Ohashi, C. T., Odermatt, B.,... Hengartner, H. (1991). Ablation of "tolerance" and induction of diabetes by virus infection in viral antigen transgenic mice. Cell, 65(2), 305-317. doi:10.1016/0092-8674(91)90164-t

Williams, M. A., Tyznik, A. J., and Bevan, M. J. (2006). Interleukin-2 signals during priming are required for secondary expansion of CD8^+^ memory T cells. Nature, 441(7095), 890-893.

doi:10.1038/nature04790